# Designing Less Forgetful Networks for Continual Learning

## Abstract

Neural networks usually excel in learning a single task. Their weights are plastic and help them to learn quickly, but these weights are also known to be unstable. Hence, they may experience catastrophic forgetting and lose the ability to solve past tasks when assimilating information to solve a new task. Existing methods have mostly attempted to address this problem through *external* constraints. Replay shows the backbone network externally stored memories; regularisation imposes additional learning objectives; and dynamic architecture often introduces more parameters to host new knowledge. In contrast, we look for *internal* means to create less forgetful networks. This paper demonstrates that two simple architectural modifications – **Masked Highway Connection** and **Layer-Wise Normalisation** – can drastically reduce the forgetfulness in a backbone network. When naïvely employed to sequentially learn over multiple tasks, our modified backbones were as competitive as those unmodified backbones with continual learning techniques applied. Furthermore, our proposed architectural modifications were compatible with most if not all continual learning archetypes and therefore helped those respective techniques in achieving new state of the art[1].

## 1 Introduction

*Continual learning* (McCloskey & Cohen, 1989) is a difficult experimental setup for most neural networks. It places a single learning agent in a dynamic environment where it must learn to adapt from a stream of tasks to make new predictions. This setup means that the underlying data are *not independent and identically distributed* (non-iid). Thus when we use conventional gradient descent methods (Rumelhart et al., 1986) to fine-tune a pre-trained network for a new task, its weights will be modified to satisfy the learning objectives of the new task. As a result, the re-parameterised network will experience *catastrophic forgetting* (McCloskey & Cohen, 1989; French, 1999) and abruptly lose the ability to solve a previously learnt task.

Recently, continual learning has been of increasing interest. Most of the latest studies can be roughly categorised as the three archetypes of *replay*, *regularisation*, and *dynamic architecture*. Replay techniques such as *gradient episodic memory* (Lopez-Paz & Ranzato, 2017) and *experience replay* (Chaudhry et al., 2019) externally store a handful of data per task. Then upon sequentially learning a new task, they re-expose copies of the stored data to the backbone network along those data of the new task. Regularisation methods like *learning without forgetting* (Li & Hoiem, 2017) and *elastic weight consolidation* (Kirkpatrick et al., 2017), on the other hand, impose the backbone network with secondary learning objectives to learn robust mappings that are less prone to parametric corruptions. Furthermore, the dynamic architectural approaches of *dynamically expandable networks* (Yoon et al., 2018) and *hard attention to the task* (Serra et al., 2018) either introduce more parameters to host new knowledge or explicitly control the backbone synaptic connections to perform inference on a subset of pre-selected weights. Though the three archetypes of continual learning methods may appear to be ideologically different, they all seek *external* means to retain learnt knowledge. In contrast, this paper will introduce *internal* modifications to mitigate forgetting.

That is, we aim to create an extremely plastic yet stable network architecture that can withstand catastrophic forgetting even when it is *naïvely implemented* (simply fine-tuned) for sequential learning over multiple tasks. Our study was inspired by the recent work of Kuo et al. (2021b). In that

---

[1] The codes will be made publicly available once the paper is published.

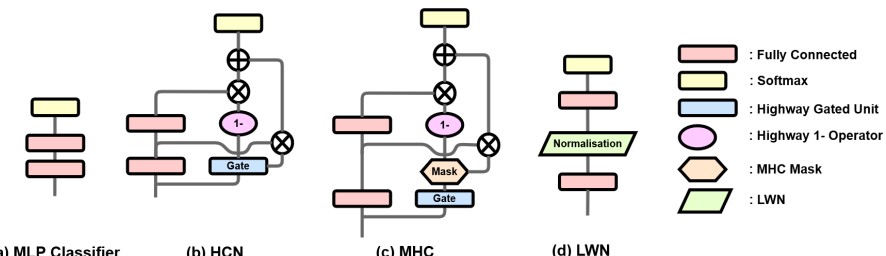

Figure 1: Different Classifier Network Designs

paper, the authors compartmentalised their backbone network. They employed a fixed feature extractor and only fine-tuned their classifiers. Unlike the conventional *multiple layer perceptron* (MLP) classifier of Figure 1(a) used in most studies (such as Mai et al. (2020)), their classifier which they called *highway connection networks* (HCNs) had the peculiar design shown in Figure 1(b). An HCN included a two-layer perceptron with *highway connection* (Srivastava et al., 2015) followed by a softmax layer. We used red to denote fully connected layers, yellow for softmax, and blue for the HCN gated unit. They demonstrated that HCNs were more robust against forgetting than MLPs; and even showed that naïvely sequential learning with HCNs could often outperform MLP classifiers with continual learning techniques applied. Moreover, they found that highway connections were specifically required and could not be replaced with *residual connections* (He et al., 2016).

In this paper, we introduce two novel modifications to HCNs. Our new ideas draw inspiration from the *stability-plasticity dilemma* (Mermillod et al., 2013). Stability is the ability to retain learnt knowledge upon parametric corruption (Sun et al., 2021); whereas plasticity refers to the ease of assimilating new information. Our study focuses on designing new architectures in the forward pass which indirectly but positively affect the updates in the backward pass. This kind of approach was previously investigated in *weight normalisation* (Salimans & Kingma, 2016). In that paper, the authors found that by reparameterising the weight vectors, they could influence the gradient covariance matrix by making it closer to an identity matrix thereby hastening network configuration.

We propose the **Masked Highway Connection** (MHC) and **Layer-Wise Normalisation** (LWN) shown in Figures 1(c) and 1(d) respectively. While similar techniques exist for general machine learning; their applications for continual learning are not well studied. After discussing how HCN mitigates forgetting (Section 3), we will introduce our MHC and show that HCNs can be made even more robust via self-identifying important weights that are worth fine-tuning through a masking mechanism (Section 4). Then, we will alternatively characterise forgetting as shifts that occur in previous distributions (Ebrahimi et al., 2020); and that the LWN can reduce such shifts (Section 5).

Our experiments (Section 6) indicated that both MHC and LWN were robust against forgetting and greatly outperformed their HCN and MLP baselines. In addition, both designs were compatible with the continual learning archetypes of replay, regularisation, and dynamic architecture; and they were able to help those respective techniques to achieve new state of the art. Moreover, MHC and LWN could also be jointly implemented. Together, they often produced even more robust results and this was because they each addressed a different aspect of catastrophic forgetting. In order to suggest the best practice for MHC and LWN for continual learning, we also tested our novel classifiers over several different settings. Importantly, we found that LWN should not be incautiously substituted with *batch normalisation* (Ioffe & Szegedy, 2015).

## 2   THE CONTINUAL LEARNING FRAMEWORK

As stated in Lopez-Paz & Ranzato (2017), the continual learning setup exposes a base learner $F$ to a continuum of data with $n$ locally, but not globally, iid tasks $\omega$ where

$$(x_1^1, \omega_1, y_1^1), \ldots, (x_\phi^i, \omega_i, y_\phi^i), \ldots, (x_\Phi^n, \omega_n, y_\Phi^n). \tag{1}$$

Each task corresponds to a dataset $(x_\phi^i, y_\phi^i) \in \mathcal{D}_{\omega_i}$ with $\Phi$ samples $\phi = 1, \ldots, \Phi$. Once the learner $F$ observes data from the $j$-th task, full access to an earlier dataset $\omega_i$ where $i < j$ is prohibited to consolidate learnt knowledge. The performance is tracked using a matrix $\mathscr{A} \in \mathbb{R}^{(n+1) \times n}$ where each column represents a distinct task. When $F$ is initialised, its accuracy over all tasks is recorded in row $\mathscr{A}_{1,(1:n)}$. After learning the $i$-th task, the new performances are updated in row $\mathscr{A}_{(i+1),(1:n)}$.

## 3 HIGHWAY CONNECTIONS IMPLICITLY RE-SCALE GRADIENTS

An intermediate layer $L_\gamma$ in an MLP can be represented as

$$\mathbf{a}_{L_\gamma} = \mathscr{H}(\mathbf{a}_{L_{(\gamma-1)}}, \mathbf{W}_{L_\gamma}) \tag{2}$$

where the layer output $\mathbf{a}_{L_\gamma}$ is transformed from the layer input $\mathbf{a}_{L_{\gamma-1}}$ over the non-linear function $\mathscr{H}$ with weight $\mathbf{W}_{L_\gamma}$. In addition, say there are $\Gamma$ layers then the network output is $a_{L_\Gamma} = \hat{\mathbf{y}}$. To fine-tune $\mathbf{W}_{L_\gamma}$, we use gradient descent with the derivative of a cost $C$ with respect to the parameters

$$\frac{\partial C}{\partial \mathbf{W}_{L_\gamma}} = \frac{\partial C}{\partial \hat{\mathbf{y}}} \underbrace{\left[ \prod_{\zeta=\gamma}^{\Gamma-1} \frac{\partial \mathbf{a}_{L_{(\zeta+1)}}}{\partial \mathbf{a}_{L_\zeta}} \right]}_{\text{Rel: Srivastava et al. (2015)}} \overbrace{\left\{ \frac{\partial \mathbf{a}_{L_\gamma}}{\partial \mathbf{W}_{L_\gamma}} \right\}}^{\text{Rel: Kuo et al. (2021b)}} . \tag{3}$$

Below, we discuss how gradient vanishing and catastrophic forgetting are closely related (Rel:).

Deep MLPs are known to suffer *gradient vanishing* (Bengio et al., 1994). When the layers increase but the magnitudes of the individual Jacobian terms $\frac{\partial \mathbf{a}_{L_{(\zeta+1)}}}{\partial \mathbf{a}_{L_\zeta}}$ are small (*i.e.,* $\in [0, 1]$), the term $\prod_{\zeta=\gamma}^{\Gamma-1} \frac{\partial \mathbf{a}_{L_{(\zeta+1)}}}{\partial \mathbf{a}_{L_\zeta}}$ diminishes to 0 (and so does $\frac{\partial C}{\partial \mathbf{W}_{L_\gamma}}$) and no effective update is made to $\mathbf{W}_{L_\gamma}$.

Highway connection (Srivastava et al., 2015) is well-known for its ability to withstand gradient vanishing. It reformulates an intermediate layer as

$$\mathbf{k}_{L_\gamma} = (1 - \mathbf{T}_{L_\gamma}) \odot \mathbf{k}_{L_{\gamma-1}} + \mathbf{T}_{L_\gamma} \odot \mathscr{H}(\mathbf{k}_{L_{(\gamma-1)}}, \mathbf{W}_{L_\gamma}) \text{ with} \tag{4}$$

$$\text{gate } \mathbf{T}_{L_\gamma} = \mathbf{T}(\mathbf{k}_{L_{(\gamma-1)}}, \mathbf{Q}_{\mathbf{L}_\gamma}) \tag{5}$$

and the element-wise dot product $\odot$. The colour-coded gated unit $\mathbf{T}_{L_\gamma}$ has its own weight $\mathbf{Q}_{\mathbf{L}_\gamma}$ and its values lie in the range of $[0, 1]$. This new formulation has the Jacobian term

$$\frac{\partial \mathbf{k}_{L_\gamma}}{\partial \mathbf{k}_{L_{(\gamma-1)}}} = \begin{cases} \mathbb{I}, & \text{if } \mathbf{T}_{L_\gamma} = \mathbf{0} \\ \mathscr{H}'(\mathbf{k}_{L_{(\gamma-1)}}, \mathbf{W}_{L_\gamma}), & \text{if } \mathbf{T}_{L_\gamma} = \mathbf{1} \end{cases} . \tag{6}$$

By introducing the identity matrix $\mathbb{I}$ to the Jacobian (and thus the [.] term of Equation (3)), highway connections prevent the vanishing of gradients and thus facilitate the training of deep networks.

In Kuo et al. (2021b), the authors noted a second property unique to highway connections. While the $\{.\}$ term of Equation (3) of an MLP was $\frac{\partial \mathbf{a}_{L_\gamma}}{\partial \mathbf{W}_{L_\gamma}} = \frac{\partial \mathscr{H}(\mathbf{a}_{L_{(\gamma-1)}}, \mathbf{W}_{L_\gamma})}{\partial \mathbf{W}_{L_\gamma}}$, a highway connection had the alternative form of $\frac{\partial \mathbf{k}_{L_\gamma}}{\partial \mathbf{W}_{L_\gamma}} = \mathbf{T}_{L_\gamma} \odot \frac{\partial \mathscr{H}(\mathbf{k}_{L_{(\gamma-1)}}, \mathbf{W}_{L_\gamma})}{\partial \mathbf{W}_{L_\gamma}}$ which included the gated unit. Since each entry in $\mathbf{T}_{L_\gamma}$ lied in $[0, 1]$, highway connections thus implicitly scaled down the magnitudes of the gradients used in gradient descent. This was desirable for continual learning because less changes were made to a pre-trained weight $\mathbf{W}_{L_\gamma}$; and HCNs were hence less susceptible to forgetting.

## 4 IMPROVING THE HCN ARCHITECTURE

Though highway connection is effective against catastrophic forgetting, its best practices and short-comings remain unclear. In this section, we review the different roles that the gated unit $\mathbf{T}_{L_\gamma}$ plays in the forward pass (inference) and in the backward pass (update). Furthermore, we will show that there exists an interesting structural connection between the HCN gated unit $\mathbf{T}_{L_\gamma}$ and the derivative $\frac{\partial \mathscr{H}(\mathbf{k}_{L_{(\gamma-1)}}, \mathbf{W}_{L_\gamma})}{\partial \mathbf{W}_{L_\gamma}}$ which can be exploited to make the network even more robust to forgetting.

Overleaf in the top half of Figure 2, we show that HCN performs a *point-to-point* gradient reduction. On the left, we superimpose a dotted box on top of HCN. The inference and update that occur within this box are illustrated on the right. Without loss of generality, the scenario in the figure is for 3 hidden dimensions and that each $\mathbf{T}_{L_\gamma}$ entry scales down its corresponding feature of the fully connected layer output $\mathbf{k}_{L_\gamma}$ during inference. Note that the entries of $\mathbf{k}_{L_\gamma}$ are shown in different shades of red; and they correspond to specific rows in the weight $\mathbf{W}_{L_\gamma}$. Likewise for update, we highlight the rows of $\frac{\partial \mathscr{H}(\mathbf{k}_{L_{(\gamma-1)}}, \mathbf{W}_{L_\gamma})}{\partial \mathbf{W}_{L_\gamma}}$ in different shades of green; and we show how each row of this derivative is scaled down, in a point-to-point fashion, by the entries of the gated unit $\mathbf{T}_{L_\gamma}$.

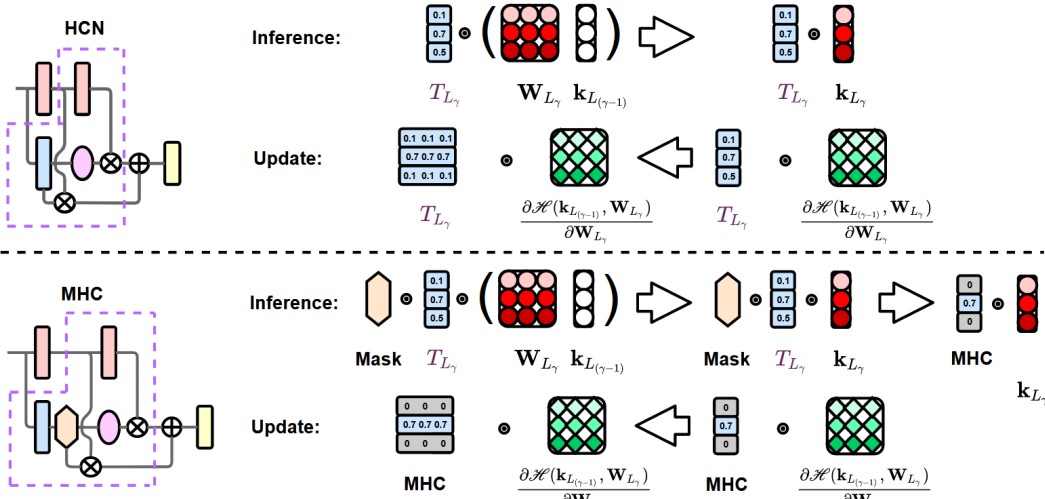

Figure 2: Differences in the Inferences and Updates of the HCN and MHC Classifiers

But does the entirety of weight $\mathbf{W}_{L_\gamma}$ need to be updated at all times? Two prior studies, the *lottery ticket hypothesis* (LTH) (Frankle & Carbin, 2019) and the *uncertainty-regularised continual learning* (UCL) (Ahn et al., 2019), suggested not. In network pruning, LTH found that only a small subset of the dense network needed to be updated to achieve competitive performances. Whereas UCL showed that continual learning could be achieved by regularising specific nodes in the weight that deviated from a pre-determined distribution. They both considered *the structural relationship* among nodes in the base learner. A similar approach can be applied to HCN.

The lower-half of Figure 2 presents the inner workings of our **Masked Highway Connecton** (MHC) classifier. We add a binary Mask to HCN and change the formulation from Equations (4) and (5) to

$$\mathbf{k}_{L_\gamma} = (1 - \mathbf{G}_{L_\gamma}) \odot \mathbf{k}_{L_\gamma - 1} + \mathbf{G}_{L_\gamma} \odot \mathscr{H}(\mathbf{k}_{L_{(\gamma-1)}}, \mathbf{W}_{L_\gamma}) \text{ with} \quad (7)$$

$$\text{a masked gate } \mathbf{G}_{L_\gamma} = \text{Mask} \odot \mathbf{T}_{L_\gamma} = \text{Mask} \odot \mathbf{T}(\mathbf{k}_{L_{(\gamma-1)}}, \mathbf{Q}_{\mathbf{L}_\gamma}). \quad (8)$$

The Mask keeps the largest neural values in $\mathbf{T}_{L_\gamma}$ with the top-$\mathscr{K}$ function. As shown in Figure 2, when $\mathbf{T}_{L_\gamma} \in \mathbb{R}^3$ and that $\mathscr{K} = 1$, the MHC masked gate $\mathbf{G}_{L_\gamma}$ nullifies the two rows with weaker activation values. The mask also nullifies specific updates during optimisation. In our experiments in Section 6, we will demonstrate that this minimalistic update can reduce excessive weight modification thereby improving network stability for the sequential learning setup.

## 5 FORGETTING, MEASURED THROUGH FEATURE SHIFTS

The gating mechanism in HCN showed that Kuo et al. (2021b) regarded forgetting as the amount of change that occurred in the pre-trained weights. However, both Ebrahimi et al. (2020) and Ahn et al. (2019) quantified forgetting through alternative means. Both studies employed *Bayesian neural networks* (BNNs) (Mackay, 1992) for continual learning. The former conceptualised forgetting as shifts that occurred in the distributions of weights; and the latter showed that BNNs could become stable by regularising the network with changes in the standard deviations of the weight matrix.

Thus in addition to MHC, we further propose **Layer-Wise Normalisation** (LWN) to mitigate shifts in the classifier. However, we address *internal covariate shifts* (ICSs) (Shimodaira, 2000) instead of weight distribution shifts. ICS refers to the change in network activation distributions due to parametric changes during training; and we will show that forgetting can also be mitigated at the *feature level*. An LWN with hidden dimension $H$ changes Equation 2 to a

$$\text{normalised layer} \quad \mathbf{a}_{L_\gamma} = \frac{\mathbf{a}_{L_\gamma}^* - \mu_{L_\gamma}}{\sigma_{L_\gamma}} \quad \text{from} \quad \mathbf{a}_{L_\gamma}^* = \mathscr{H}(\mathbf{a}_{L_{(\gamma-1)}}, \mathbf{W}_{L_\gamma}) \text{ with} \quad (9)$$

$$\text{mean:} \quad \mu_{L_\gamma} = \frac{1}{H}\sum_{\eta=1}^{H} \mathbf{a}_{L_\gamma, \eta}^* \quad \text{and} \quad \text{standard deviation:} \quad \sigma_{L_\gamma} = \sqrt{\frac{1}{H}\sum_{\eta=1}^{H}(\mathbf{a}_{L_\gamma, \eta}^* - \mu_{L_\gamma})^2}. \quad (10)$$

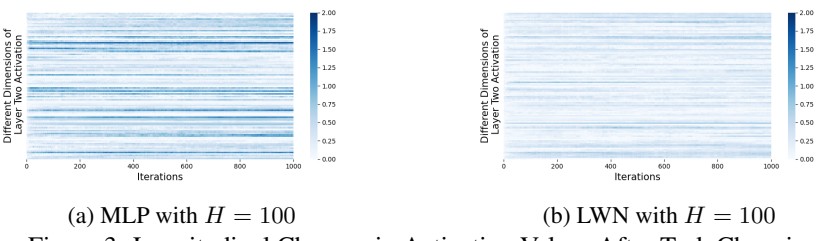

(a) MLP with $H = 100$                (b) LWN with $H = 100$

Figure 3: Longitudinal Changes in Activation Values After Task Changing

In order to demonstrate the effectiveness of LWNs over MLPs, we prepared a toy example which first trained both classifiers with MNIST (LeCun et al., 1998) and then with FashionMNIST (Xiao et al., 2017). Both classifiers had the hidden dimensions $H = 100$ with the two layer designs shown in Figures 1(d) and 1(a) respectively. Both datasets contained grey-scale images on $28 \times 28$ pixel boxes with 60K and 10K images for training and testing. After pre-training both classifiers on MNIST, we externally stored a batch of MNIST images $B_{\mathrm{MNIST}}$ and applied them on the classifiers to collect the output activations of the second layer $\mathbf{a}_{L_2}^{(0)}$. When we fine-tuned the classifiers for FashionMNIST, we re-applied $B_{\mathrm{MNIST}}$ and collected the new activation $\mathbf{a}_{L_2}^{(\xi)}$ for each $\xi$-th update. We then plotted the changes in activation in Figures 3(a) and 3(b). See Appendix A for more details in the setup.

The changes in activation were defined as

$$\mathrm{Diff}_{\mathrm{act}} = \mathrm{abs}\left( \mathbb{E}_{b=1}^{|B_{\mathrm{MNIST}}|} \left( {}_b\mathbf{a}_{L_2,\eta}^{(\xi)} - {}_b\mathbf{a}_{L_2,\eta}^{(0)} \right) \right) \tag{11}$$

and it measured the magnitudes of the expected differences in the changes of the layer two activation (over samples $b$ in $B_{\mathrm{MNIST}}$). The deeper the colours the larger the differences. Each row represented the changes in a unique feature $\mathbf{a}_{L_2,\eta}$ after the classifier received updates to solve FashionMNIST. Each row generally became deeper and this showed that ICS occurred during re-parameterisation and that the learnt mapping for MNIST images $B_{\mathrm{MNIST}}$ had diverged. After fine-tuning, we found that the divergence was $\mathrm{Diff}_{\mathrm{act}} = 86.44$ in MLP and $\mathrm{Diff}_{\mathrm{act}} = 69.03$ in LWN. Forgetting was hence $20\%(= [1 - 69.03/86.44] \times 100\%)$ less severe in LWN than in MLP. This was because that the differences in LWN features became smaller after redistributing the activation along the feature dimension. LWN also scaled well with larger dimensionality; see Appendix B for details.

Unlike the well-known *layer normalisation* (Ba et al., 2016), our LWN did not have learnable parameters to reshape the post-normalised activation distribution. This was because that during our experiments, we found that these features were not necessary. Also, we found that LWN was specifically required to mitigate forgetting; and it could not be replaced by the popular batch normalisation (Ioffe & Szegedy, 2015). This will be further discussed in Section 6.3.

## 6 EXPERIMENTS AND RESULTS

We compared six classifier networks with four continual learning setups over three datasets. The main results were summarised in Tables 1 and 2, with more results in Tables 4 – 12 in the appendices.

**The Datasets**
We tested image permutation (Kirkpatrick et al., 2017) on MNIST (LeCun et al., 1998), and incremental classes (Rebuffi et al., 2017) on Cifar100 (Krizhevsky, 2009) and CUB200 (Wah et al., 2011). MNIST was previously mentioned in Section 5. Cifar100 included 100 classes of $32 \times 32$ coloured images; with 50K for training and 10K for testing. Whereas CUB200 contained $5,994$ training and $5,794$ test images over 200 coloured bird species on mostly $500 \times 500$ pixel boxes.

**The Tasks**
Permuted MNIST (Perm-MNIST) was derived by first reshaping MNIST images as vectors of $784(= 28 \times 28)$ pixels, then by applying a unique permutation on all vectors of that task. On the other hand, incremental Cifar100 (Inc-Cifar100) and incremental CUB200 (Inc-CUB200) were generated via sequentially introducing a fixed set of classes per task drawn without replacement from the datasets. There were 20 tasks for Perm-MNIST. For Inc-Cifar100, we tested three scenarios with 5, 10, and 20 classes per task. Likewise for Inc-CUB200, we introduced 10, 20, and 40 classes per task. (The Inc scenarios thus had 20, 10, and 5 respective tasks.) We mainly followed Douillard et al. (2020)

to process the datasets[2] and setup as domain-incremental learning (Van de Ven & Tolias, 2019).

**The Backbone Networks**
For Perm-MNIST, our backbone networks were simply two-layer MLPs, HCNs, and MHCs. We tested MHCs with a range of different settings – we used `top-`$\mathscr{K}$ to keep between the top $10\%$ to the top $90\%$ of the gated unit activation. In addition, we embedded LWN in MLPs, in HCNs, and in MHCs. All of the normalisation were applied specifically to the input of layer two. Interested readers could find elaborated illustrations in Figure 7 of Appendix C.

For Inc-Cifar100 and Inc-CUB200, our backbone networks were compartmentalised as fixed pre-trained feature extractors with sequential learning classifiers. The feature extractors for Inc-Cifar100 were ResNet18s (He et al., 2016) pre-trained on ImageNet (Deng et al., 2009); and those for Inc-CUB200 were ResNeXt50s (Xie et al., 2017) pre-trained on ImageNet. We replaced the final layer of the feature extractors with our classifiers. Our classifiers here were also two-layer MLPs, HCNs, and MHCs. The MHC again kept `top-`$\mathscr{K}$ with $\mathscr{K} \in [10\%, 90\%]$; however, we embedded the LWNs differently. The normalisation were applied in two different places – first on the input of layer one (extracted by the feature extractors), and then on the input of layer two. Again, interested readers could find elaborated illustrations in Figure 8 of Appendix C.

**Additional Training Detials**
For Perm-MNIST and Inc-Cifar100, our classifiers had hidden dimensions $H = 100$; and it was $H = 400$ for Inc-CUB200. The Perm-MNIST backbones observed 5K images on Task 1 then 1K images for all subsequent tasks. Likewise for Inc-Cifar100, it was 5 epochs on Task 1 then 1 epoch for the rest. However, the backbones observed 5 epochs for all Inc-CUB200 tasks. We used batch size $B = 10$, and updated all classifiers with SGD (Rumelhart et al., 1986) with learning rate 0.01.

**The Continual Learning Setups**
We denoted *singular* as the scenario when the backbones were naïvely and sequentially trained (simply fine-tuned) over all tasks. Then, we tested the regularisation-based *elastic weight consolidation* (EWC) (Kirkpatrick et al., 2017), the memory-based *experience replay* (ER) (Chaudhry et al., 2019), and the dynamic architectural-based *hard attention to the task* (HAT) (Serra et al., 2018)[3]. Some secondary experiments could be found in Appendix F.

**Metrics**
We used the two metrics below to evaluate the sequential learning performances

| | | | |
|---|---|---|---|
| **Average accuracy** | (ACC) | $\frac{1}{n} \sum_{i=1}^{n} \mathscr{A}_{n+1,i}$ | ; and |
| **Final Accuracy of Task 1** | (FA1) | $\mathscr{A}_{n+1,1}$. | |

Both metrics were based on matrix $\mathscr{A}$ of Section 2. The classifier plasticity could be indicated through ACC, and FA1 could reflect the classifier stability. All units were in percentage accuracy, the higher the better; and all results were reported as the $95\%$ confidence intervals over 10 seeds.

### 6.1 RESULTS ON NAÏVELY SEQUENTIAL LEARNING

We summarised a subset of results for the *singular* setup overleaf in Table 1. It included all results for Perm-MNIST, Inc-Cifar100 with 20 classes per task, and Inc-CUB200 with 40 classes per task. The baseline results for MLPs and HCNs were shown in grey, and we highlighted our results that were better than the baselines in cyan. This table reflected four important remarks.

The first remark was that, our classifier designs could outperform the baselines with a large margin if they were configured properly. However the second remark showed that, the best practice of MHC varied for each task. For Perm-MNIST, the best performce in MHC was recorded when we kept the top $\mathscr{K} = 70\%$ of gated unit activation; but for Inc-Cifar100 and Inc-CUB200, it was when only the top $\mathscr{K} = 10\%$ activation was kept. This was potentially due to the contextual complexity of the datasets; such that it was possible to learn with less information from the contextually rich Cifar100 and CUB200. The third remark was that LWN greatly improved both the plasticity (ACC score) and stability (FA1 score) for all classifiers. While it was well-known that normalisation could improve network plasticity (Ba et al., 2016), our work provided empirical evidence that normalisation along the feature dimension was beneficial for mitigating forgetting. In addition, the fourth remark was

---

[2]Refer to `inclearn/lib/data/datasets.py` in Douillard et al. (2020)'s official repository.
[3]We set the regularisation coefficient $\lambda = 3$ for EWC; stored 50 images per task for ER; and applied HAT only on the first layer of classifier synapses.

Table 1: A Subset of Results on the Naïve Sequential Learning of Classifiers

| Task | Ours | Classifier | ACC (%) | FA1 (%) |
|---|---|---|---|---|
| Perm-MNIST | - - | MLP | $66.4 \pm 23.1$ | $67.9 \pm 11.9$ |
| | - - | HCN | $72.0 \pm 15.5$ | $78.7 \pm 8.4$ |
| | ✓ | MHC (90%) | $72.1 \pm 16.1$ | $79.1 \pm 7.7$ |
| | ✓ | MHC (70%) | $72.4 \pm 15.1$ | $79.5 \pm 7.3$ |
| | ✓ | MHC (50%) | $72.1 \pm 14.7$ | $79.3 \pm 7.4$ |
| | ✓ | MHC (30%) | $71.4 \pm 14.5$ | $78.4 \pm 7.3$ |
| | ✓ | MHC (10%) | $70.8 \pm 15.1$ | $76.1 \pm 8.4$ |
| | ✓ | LWN in MLP | $68.4 \pm 23.5$ | $71.3 \pm 11.5$ |
| | ✓ | LWN in HCN | $74.6 \pm 14.2$ | $80.5 \pm 7.5$ |
| | ✓ | LWN in MHC (70%) | $\mathbf{75.3 \pm 13.1}$ | $\mathbf{81.6 \pm 8.4}$ |
| Inc-Cifar100 | - - | MLP | $41.7 \pm 18.1$ | $32.1 \pm 5.9$ |
| (inc. 20 classes) | - - | HCN | $46.0 \pm 12.1$ | $41.8 \pm 3.8$ |
| | ✓ | MHC (90%) | $46.1 \pm 11.9$ | $41.2 \pm 4.0$ |
| | ✓ | MHC (70%) | $46.2 \pm 12.1$ | $40.8 \pm 5.9$ |
| | ✓ | MHC (50%) | $46.6 \pm 11.4$ | $40.8 \pm 4.6$ |
| | ✓ | MHC (30%) | $46.7 \pm 10.8$ | $41.8 \pm 4.3$ |
| | ✓ | MHC (10%) | $46.9 \pm 10.7$ | $42.0 \pm 6.1$ |
| | ✓ | LWN in MLP | $45.3 \pm 16.0$ | $37.0 \pm 8.1$ |
| | ✓ | LWN in HCN | $48.0 \pm 10.8$ | $42.7 \pm 4.2$ |
| | ✓ | LWN in MHC (10%) | $\mathbf{48.4 \pm 9.6}$ | $\mathbf{44.2 \pm 4.5}$ |
| Inc-CUB200 | - - | MLP | $55.0 \pm 19.7$ | $45.3 \pm 11.1$ |
| (inc. 40 classes) | - - | HCN | $57.4 \pm 15.3$ | $51.7 \pm 6.4$ |
| | ✓ | MHC (90%) | $56.8 \pm 15.4$ | $51.2 \pm 7.7$ |
| | ✓ | MHC (70%) | $55.2 \pm 17.1$ | $48.3 \pm 7.6$ |
| | ✓ | MHC (50%) | $56.8 \pm 15.2$ | $51.4 \pm 5.8$ |
| | ✓ | MHC (30%) | $57.5 \pm 15.7$ | $53.6 \pm 6.7$ |
| | ✓ | MHC (10%) | $60.8 \pm 14.0$ | $57.6 \pm 4.6$ |
| | ✓ | LWN in MLP | $71.7 \pm 19.1$ | $61.3 \pm 9.5$ |
| | ✓ | LWN in HCN | $75.6 \pm 16.3$ | $66.0 \pm 5.7$ |
| | ✓ | LWN in MHC (10%) | $\mathbf{76.2 \pm 15.3}$ | $\mathbf{67.3 \pm 5.3}$ |

that MHC and LWN could be made more powerful when they were jointly implemented. This was not particularly surprising because both techniques addressed forgetting from different perspectives. MHCs considered forgetting as the modifications made to the pre-trained weights (see Section 4); whereas LWNs were effective in minimising the feature shifts in a learnt mapping (see Section 5).

We made more results for the *singular* setup available in Appendix D. We refer to Table 4 for additional results on Inc-Cifar100; and to Table 5 for additional results on Inc-CUB200.

## 6.2 RESULTS WITH CONTINUAL LEARNING TECHNIQUES APPLIED

After the *singular* setup, we tested all classifiers with EWC, ER, and HAT applied. Our results were summarised in Table 2 overleaf. Due to space constraints, we selected the best performing LWN in MHC results. Interested readers could find the results for all other backbone network designs in Tables 6 – 12 in Appendix D. We found three additional important remarks from the experiments.

The first remark was that our LWN in MHCs were compatible with all of EWC, ER, and HAT. Since these three techniques were from the most well-known continual learning archetypes, we expected our LWN in MHCs to be compatible with most if not all existing continual learning techniques. Second, our LWN in MHCs achieved better results than the baselines. Hence, they helped the aforementioned techniques in reaching new state-of-the-art status. The third remark was that, the majority of our naïvely sequential learning LWN in MHCs from Table 1 (both ACC and FA1) achieved better results than the baseline classifiers with continual learning techniques applied. Hence, we demonstrated the importance of employing the correct classifier design to mitigate forgetting.

## 6.3 BATCH NORMALISATION IS NOT GOOD FOR CONTINUAL LEARNING

In this section, we tested *batch normalisation* (BN) (Ioffe & Szegedy, 2015) in MLP. The results for Perm-MNIST in Table 3 showed that BN was detrimental for the backbone networks.

Table 2: A Subset of Results with Continual Learning Techniques Applied

| Task | Ours | CL Tech. | Classifier | ACC (%) | FA1 (%) |
|---|---|---|---|---|---|
| Perm-MNIST | - - | EWC | MLP | $68.9 \pm 18.3$ | $76.5 \pm 7.0$ |
| | - - | EWC | HCN | $73.7 \pm 14.4$ | $82.3 \pm 5.7$ |
| | ✓ | EWC | LWN in MHC (70%) | $\mathbf{76.5 \pm 13.3}$ | $\mathbf{84.6 \pm 6.2}$ |
| Perm-MNIST | - - | ER | MLP | $72.1 \pm 12.5$ | $80.3 \pm 2.8$ |
| | - - | ER | HCN | $73.3 \pm 15.6$ | $83.9 \pm 2.1$ |
| | ✓ | ER | LWN in MHC (70%) | $\mathbf{76.0 \pm 14.3}$ | $\mathbf{85.9 \pm 1.9}$ |
| Perm-MNIST | - - | HAT | MLP | $67.2 \pm 20.6$ | $72.3 \pm 18.5$ |
| | - - | HAT | HCN | $71.6 \pm 15.1$ | $79.2 \pm 6.0$ |
| | ✓ | HAT | LWN in MHC (70%) | $\mathbf{75.3 \pm 13.5}$ | $\mathbf{83.2 \pm 4.0}$ |
| Inc-Cifar100 | - - | EWC | MLP | $43.3 \pm 15.2$ | $36.1 \pm 5.5$ |
| (inc. 20 classes) | - - | EWC | HCN | $47.0 \pm 10.6$ | $44.0 \pm 4.9$ |
| | ✓ | EWC | LWN in MHC (10%) | $\mathbf{49.2 \pm 8.6}$ | $\mathbf{46.9 \pm 3.2}$ |
| Inc-Cifar100 | - - | ER | MLP | $41.7 \pm 17.0$ | $37.1 \pm 5.5$ |
| (inc. 20 classes) | - - | ER | HCN | $42.1 \pm 16.2$ | $38.5 \pm 5.2$ |
| | ✓ | ER | LWN in MHC (10%) | $\mathbf{45.1 \pm 14.4}$ | $\mathbf{41.6 \pm 4.8}$ |
| Inc-Cifar100 | - - | HAT | MLP | $43.4 \pm 13.0$ | $38.0 \pm 6.0$ |
| (inc. 20 classes) | - - | HAT | HCN | $44.2 \pm 13.1$ | $38.6 \pm 8.3$ |
| | ✓ | HAT | LWN in MHC (10%) | $\mathbf{47.3 \pm 9.1}$ | $\mathbf{44.9 \pm 4.8}$ |
| Inc-CUB200 | - - | EWC | MLP | $56.0 \pm 17.8$ | $46.8 \pm 13.5$ |
| (inc. 40 classes) | - - | EWC | HCN | $58.8 \pm 13.7$ | $51.8 \pm 6.6$ |
| | ✓ | EWC | LWN in MHC (10%) | $\mathbf{77.4 \pm 14.0}$ | $\mathbf{69.7 \pm 7.7}$ |
| Inc-CUB200 | - - | ER | MLP | $51.9 \pm 20.1$ | $47.6 \pm 4.5$ |
| (inc. 40 classes) | - - | ER | HCN | $52.0 \pm 20.2$ | $48.2 \pm 9.0$ |
| | ✓ | ER | LWN in MHC (10%) | $\mathbf{71.8 \pm 20.0}$ | $\mathbf{63.1 \pm 8.6}$ |
| Inc-CUB200 | - - | HAT | MLP | $55.7 \pm 16.1$ | $59.6 \pm 6.1$ |
| (inc. 40 classes) | - - | HAT | HCN | $56.4 \pm 14.1$ | $62.1 \pm 7.4$ |
| | ✓ | HAT | LWN in MHC (10%) | $\mathbf{78.8 \pm 7.7}$ | $\mathbf{77.5 \pm 4.6}$ |

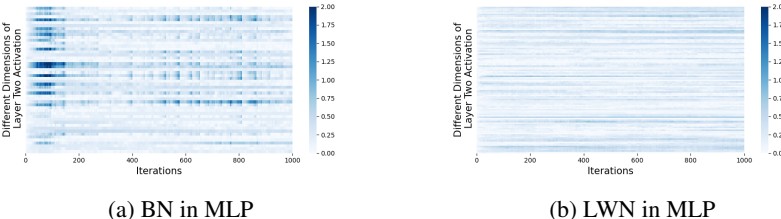

(a) BN in MLP       (b) LWN in MLP

Figure 4: Longitudinal Changes in Activation Values After Task Changing

In order to understand the reason why BN was disadvantageous for continual learning, we repeated the experiments in Section 5 where we first trained BN in MLP for MNIST and then fine-tuned for FashionMNIST. We illustrated the results in Figure 4. This figure was formatted following the style of Figure 3. BN in MLP had much

Table 3: Results on Perm-MNIST

| Classifier | ACC (%) | FA1 (%) |
|---|---|---|
| MLP | 66.4 | 67.9 |
| LWN in MLP | **68.4** | **71.3** |
| BN in MLP | 44.3 | 38.3 |

deeper colours than LWN in MLP. In addition, the divergence was $\text{Diff}_{\text{act}} = 97.66$ in BN in MLP and forgetting was hence $41\%(= [1 - 69.03/97.66] \times 100\%)$ more severe than LWN in MLP.

The deep colours first appeared after the BN in MLP started to observe data from FashionMNIST. This indicated that the learnt parameters (*i.e.,* the *scale* and the *shift*) in BN required readjustments as a response to the change in source data distribution. This was expected since BN normalised neural network values along the batch dimension rather than the feature dimension. That is, while BN could decrease the ICS within a single task, it could not decrease the ICS occurring between tasks. These results showed that LWN was specifically required and that it should not be incautiously replaced with an alternative normalisation.

### 6.4 SEQUENTIAL LEARNING IN A NOISY ENVIRONMENT

We further concerned the realistic scenario of continual learning under the presence of noise. The *noise injection* (NI) (Kuo et al., 2021a) scenario randomly shuffled a portion of pixels of Perm-MNIST images. The lowest setting injected 10% noise while the highest setting injected 50% noise. Refer to the details of this setup in Appendix E.

We naïvely sequentially trained the classifiers and plotted the results in Figure 5. Both HCNs and LWN in MHCs were able to achieve better results than MLPs on plasticity (ACC). In addition, our novel LWN in MHCs also outperformed HCNs in stability (FA1). We attributed this to LWN in MHC's minimalistic updates for reducing the impact of input corruption on optimisation.

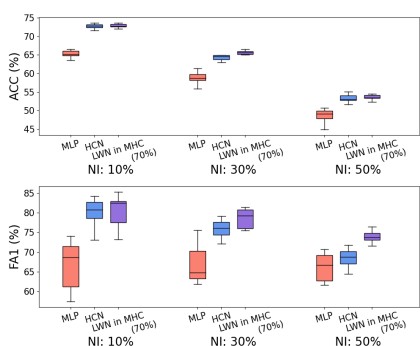

Figure 5: Perm-MNIST with Noise

## 7 RELATED WORK

Our results highlight the architectural importance for continual learning. The components in the forward pass not only impact plasticity, but also influence the backward pass and thus stability. From this perspective, continual learning shares many important research questions with *network pruning* and *neural architectural search* (NAS).

Network pruning was originally proposed to reduce both the computational and memory cost of neural networks. Han et al. (2015) found that large networks could perform with no loss in accuracy with only 10% of its total parameters. Zhou et al. (2019) further demonstrated that neural networks could still perform competitively when the weights were reset but with their original signs kept. This implied that the architecture had a direct influence on the topology of the solution space (Li et al., 2018) and potentially also the capacity of the minima on the loss landscape (Hochreiter & Schmidhuber, 1997). This in turn decided how easy it was to not forget. These insights echoed with some findings in NAS. NAS considered network building as a learning problem (Zoph & Le, 2017). It was recently extended to continual learning in Mundt et al. (2021); and some linear classifiers with random weights were found to perform on par with fully trained deep counterparts.

Besides network pruning and NAS, many work in the lottery ticket hypothesis (LTH) (Frankle & Carbin, 2019) also highlighted the importance of controlled and minimalistic updates. Gohil et al. (2019) found that when configured correctly, a small and sparsified subnetwork within a dense net could be generalised across tasks. Additional studies on the structural relations between subnetworks and feature distributions were also investigated in Ayinde et al. (2019) and Kuo et al. (2021a).

## 8 CONCLUSION

This paper introduced **Masked Highway Connections** (MHCs) and **Layer-Wise Normalisations** (LWNs) as classifiers to mitigate forgetting for backbone networks that sequentially learn over several tasks. Unlike the *reply*, *regularisation*, and *dynamic architectural* continual learning archetypes, our approach did not add *external* constraints to prevent forgetting. Instead, we focused on *internal* modifications that were beneficial for the classifier design. To elaborate, our methods were able to stop forgetting without the externally stored data of replay and were thus memory efficient. Similarly, our methods were different to dynamic architectures and did not introduce new modules upon learning a new task. Hence, our methods were also computationally efficient.

We demonstrated that forgetting could be prevented at both the *weight* level and at the *feature* level. MHC achieved the former via minimalistic updates to lower the impact of weight corruption. While LWN achieved the latter through feature redistribution to retain previously learnt mappings.

MHCs and LWNs were able to outperform their baselines. In addition, they were compatible with techniques of all continual learning archetypes hence helping those techniques in achieving new state of the art. Moreover, LWN in MHCs were very stable even when they were the naïvely implemented; they often achieved better results than baseline classifiers with continual learning techniques applied.

ETHICS STATEMENT

This paper investigated simple yet effective classifier designs to prevent catastrophic forgetting in the continual learning setup. All of the experiments conducted in this paper were still on the nicely crafted datasets of MNIST, Cifar100, and CUB200. Thus, the content of this work does not present any foreseeable societal consequences.

REPRODUCIBILITY STATEMENT

All of the important hyper-parametric settings were described in text. Interested readers should refer to **The Backbone Networks** and **Additional Training Details** in Section 6 for more details. In addition, the authors of this manuscript promise to make the codes of this paper publicly available when the paper is published.

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

## A    DETAILS ON THE EXPERIMENTAL SETUP OF SECTION 5

This section aims to provide more details on the experimental setup for the toy experiment in Section 5. The purpose of the experiment is to compare the changes in the learnt mapping between an MLP classifier and an LWN in MLP classifier when we change from training them on MNIST (LeCun et al., 1998) to FashionMNIST (Xiao et al., 2017).

The design of the baseline MLP classifier looked like that in Figure 1(a), and similarly the LWN in MLP classifier took the shape of that in Figure 1(d). Both classifiers had 2 layers – with input dimension 784, hidden dimension 100, and output dimension 10; and they were trained with SGD (Rumelhart et al., 1986) with learning rate 0.01. As shown in the figure, the normalisation in LWN in MLP was applied at the input of layer two (equivalently, the output of layer one).

The two classifiers were first trained on MNIST. They were exposed to 5K copies of MNIST images in batches of size 10. Prior to input, each image was flattened as a vector of size $784(=28 \times 28)$. After learning from the 5K copies, we externally stored a subset of MNIST images of size 50 (*i.e.,* , $|B_{\text{MNIST}}| = 50$). Then, we applied the saved images $B_{\text{MNIST}}$ to the pre-trained classifiers to collect the layer two activation $\mathbf{a}_{L_2}^{(0)}$.

We proceeded to fine-tune the classifiers on 10K images of FashionMNIST. Similar to the previous setup, each image was flattened as vectors of 784 pixels and that the batch size was 10. There were hence 1K updates. After each step of update, we re-applied $B_{\text{MNIST}}$ to the fine-tuned classifiers and collected the new activation values at layer two $\mathbf{a}_{L_2}^{(\xi)}$. We then recorded the longitudinal changes in the learnt mapping for MNIST as abs $\left( \mathbb{E}_{b=1}^{|B_{\text{MNIST}}|} \left( {}_b\mathbf{a}_{L_2,\eta}^{(\xi)} - {}_b \mathbf{a}_{L_2,\eta}^{(0)} \right) \right)$ and plot the results in Figure 3.

This experiment was later re-complied in Section 6.3. The BN in MLP classifier also had the normalisation applied at the input of layer two.

## B    LWN SCALES WELL WITH LARGE HIDDEN DIMENSIONS

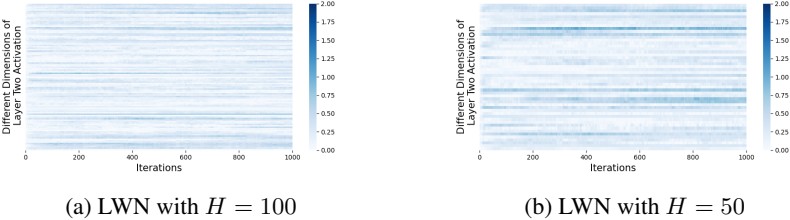

(a) LWN with $H = 100$        (b) LWN with $H = 50$

Figure 6: Longitudinal Changes in Activation Values After Task Changing

Following the experimental setup as described in Appendix A, we further compared LWNs with 100 and with 50 hidden dimensions. As shown in Figure 6, there were more patches of deeper colours in an LWN with 50 hidden dimensions than there were in an LWN with 100 hidden dimensions. After fine-tuning both classifiers, we found that an LWN with 100 hidden dimensions had a divergence score of $\text{Diff}_{\text{act}} = 69.03$ while an LWN with 50 hidden dimensions had $\text{Diff}_{\text{act}} = 37.34$. After weighting it equally across the number of dimensions, it was $\text{Diff}_{\text{act}} = 0.69 (= 69.03/100)$ per dimension for the former and $\text{Diff}_{\text{act}} = 0.75 (= 37.34/50)$ per dimension for the latter. Forgetting was hence $8.00\% (= [1 - 0.69/0.75] \times 100\%)$ less severe in the former setting. This was because that LWN exploited the hidden dimension in neural networks – the higher the dimensionality, the more evenly the features could be redistributed.

## C  FIGURATIVE CLARIFICATIONS ON THE CLASSIFIERS IN SECTION 6

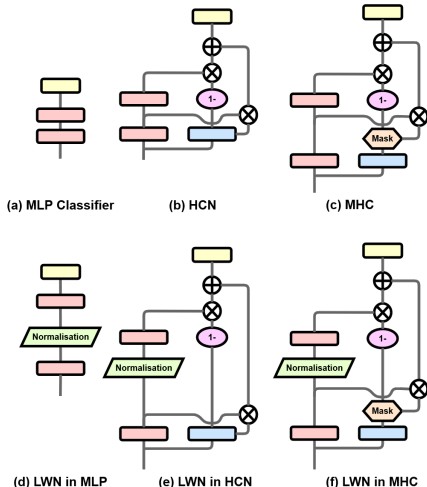

Figure 7: Different Classifier Network Designs for Perm-MNIST

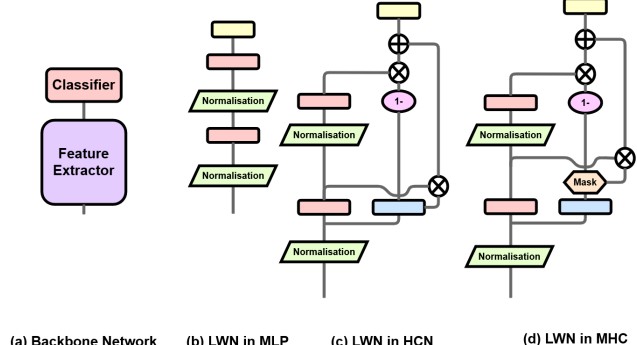

Figure 8: Different Classifier Network Designs for Incremental Classes

In Figure 7, we illustrated the different classifier designs that we used for the Perm-MNIST experiments. As mentioned previously, the backbone for the Perm-MNIST experiments were solely the classifiers and that no feature extractor was involved. In addition, all LWN were applied at the input of layer two.

Likewise, we presented our different classifier designs for Inc-Cifar100 and Inc-CUB200 overleaf in Figure 8. There were two main differences. First, the backbone networks of the incremental classes were compartmentalised as fixed pre-trained feature extractors and sequential learning classifiers. Second, the normalisation from LWN were applied at two places instead of one. In addition to the usual normalisation for the input of layer two, we added LWN to the input of layer one. That is, normalisation was applied directly after the feature extractors pre-processed the input data.

## D  MORE RESULTS FROM THE EXPERIMENTS

Due to space constraint, we were unable to put all experimental results in the main text. Hence we only showed the most important results, but we stored the rest of the performances in this section.

In Table 4, we provided additional results on Inc-Cifar100 under the naïve sequential learning setup. The results were for the scenarios with class increments of size 5 and size 10. Similarly, Table 5 provided additional results on Inc-CUB200 under the naïve sequential learning setup with class incremental size 10 and size 20.

Then in Tables 6 to 12, we listed more results when the classifiers were trained with continual learning techniques applied. The aditional results in this section provided the extra metric scores for

MHC, LWN in MLP, and LWN in HCN for all scenarios. In addition, we also listed those results for Inc-Cifar100 with incremental class sizes 5 and 10; and similarly Inc-CUB200 with incremental class sizes 10 and 20.

Table 4: More Results on the Naïve Sequential Learning of Classifiers

| Task | Ours | Classifier | ACC (%) | FA1 (%) |
|---|---|---|---|---|
| Inc-Cifar100 | - - | MLP | $63.0 \pm 38.2$ | $53.6 \pm 16.0$ |
| (inc. 5 classes) | - - | HCN | $66.9 \pm 30.4$ | $61.6 \pm 11.0$ |
| | ✓ | MHC (90%) | $66.3 \pm 31.5$ | $61.7 \pm 12.8$ |
| | ✓ | MHC (70%) | $65.9 \pm 33.0$ | $62.1 \pm 13.4$ |
| | ✓ | MHC (50%) | $67.1 \pm 30.6$ | $62.8 \pm 11.8$ |
| | ✓ | MHC (30%) | $66.7 \pm 31.5$ | $62.8 \pm 11.1$ |
| | ✓ | MHC (10%) | $66.9 \pm 33.2$ | $65.0 \pm 7.7$ |
| | ✓ | LWN in MLP | $66.4 \pm 33.2$ | $55.8 \pm 10.0$ |
| | ✓ | LWN in HCN | $66.9 \pm 33.5$ | $62.4 \pm 5.2$ |
| | ✓ | LWN in MHC (10%) | $\mathbf{67.8 \pm 31.2}$ | $\mathbf{63.9 \pm 9.3}$ |
| Inc-Cifar100 | - - | MLP | $49.9 \pm 20.5$ | $40.0 \pm 12.0$ |
| (inc. 10 classes) | - - | HCN | $54.6 \pm 15.8$ | $47.9 \pm 8.0$ |
| | ✓ | MHC (90%) | $54.1 \pm 16.6$ | $41.7 \pm 9.1$ |
| | ✓ | MHC (70%) | $54.4 \pm 16.3$ | $48.3 \pm 8.8$ |
| | ✓ | MHC (50%) | $55.1 \pm 15.5$ | $49.2 \pm 10.4$ |
| | ✓ | MHC (30%) | $55.4 \pm 15.4$ | $48.6 \pm 11.2$ |
| | ✓ | MHC (10%) | $55.5 \pm 15.0$ | $47.6 \pm 11.7$ |
| | ✓ | LWN in MLP | $54.1 \pm 18.5$ | $44.8 \pm 6.9$ |
| | ✓ | LWN in HCN | $56.4 \pm 15.6$ | $48.8 \pm 10.0$ |
| | ✓ | LWN in MHC (10%) | $\mathbf{57.3 \pm 14.2}$ | $\mathbf{52.4 \pm 8.3}$ |

Table 5: More Results on the Naïve Sequential Learning of Classifiers

| Task | Ours | Classifier | ACC (%) | FA1 (%) |
|---|---|---|---|---|
| Inc-CUB200 | - - | MLP | $57.7 \pm 46.0$ | $36.7 \pm 20.7$ |
| (inc. 10 classes) | - - | HCN | $60.9 \pm 44.8$ | $45.1 \pm 16.5$ |
| | ✓ | MHC (90%) | $60.4 \pm 44.0$ | $43.8 \pm 15.2$ |
| | ✓ | MHC (70%) | $58.9 \pm 45.7$ | $44.0 \pm 20.8$ |
| | ✓ | MHC (50%) | $58.9 \pm 44.4$ | $44.3 \pm 21.3$ |
| | ✓ | MHC (30%) | $58.9 \pm 45.9$ | $47.7 \pm 11.1$ |
| | ✓ | MHC (10%) | $62.2 \pm 42.2$ | $52.4 \pm 16.7$ |
| | ✓ | LWN in MLP | $72.0 \pm 38.3$ | $62.4 \pm 18.4$ |
| | ✓ | LWN in HCN | $\mathbf{74.6 \pm 34.7}$ | $\mathbf{71.1 \pm 10.8}$ |
| | ✓ | LWN in MHC (10%) | $74.6 \pm 35.4$ | $70.4 \pm 13.1$ |
| Inc-CUB200 | - - | MLP | $55.6 \pm 25.1$ | $50.2 \pm 16.3$ |
| (inc. 20 classes) | - - | HCN | $59.4 \pm 26.5$ | $55.1 \pm 11.1$ |
| | ✓ | MHC (90%) | $58.5 \pm 27.0$ | $54.3 \pm 11.7$ |
| | ✓ | MHC (70%) | $56.5 \pm 27.0$ | $50.4 \pm 10.2$ |
| | ✓ | MHC (50%) | $56.0 \pm 26.1$ | $51.9 \pm 9.7$ |
| | ✓ | MHC (30%) | $57.9 \pm 26.9$ | $53.4 \pm 12.0$ |
| | ✓ | MHC (10%) | $60.5 \pm 25.9$ | $56.1 \pm 12.6$ |
| | ✓ | LWN in MLP | $71.2 \pm 21.6$ | $64.9 \pm 11.4$ |
| | ✓ | LWN in HCN | $\mathbf{73.8 \pm 19.7}$ | $68.1 \pm 11.2$ |
| | ✓ | LWN in MHC (10%) | $73.5 \pm 19.0$ | $\mathbf{68.4 \pm 6.0}$ |

Table 6: More Results with Continual Learning Techniques Applied

| Task | Ours | CL Tech. | Classifier | ACC (%) | FA1 (%) |
|---|---|---|---|---|---|
| Perm-MNIST | - - | EWC | MLP | $68.9 \pm 18.3$ | $76.5 \pm 7.0$ |
|  | - - | EWC | HCN | $73.7 \pm 14.4$ | $82.3 \pm 5.7$ |
|  | ✓ | EWC | MHC (70%) | $73.7 \pm 14.4$ | $82.8 \pm 4.6$ |
|  | ✓ | EWC | LWN in MLP | $70.8 \pm 18.8$ | $79.3 \pm 7.4$ |
|  | ✓ | EWC | LWN in HCN | $\mathbf{76.5 \pm 13.1}$ | $84.2 \pm 5.8$ |
|  | ✓ | EWC | LWN in MHC (70%) | $76.5 \pm 13.3$ | $\mathbf{84.6 \pm 6.2}$ |
| Perm-MNIST | - - | ER | MLP | $72.1 \pm 12.5$ | $80.3 \pm 2.8$ |
|  | - - | ER | HCN | $73.3 \pm 15.6$ | $83.9 \pm 2.1$ |
|  | ✓ | ER | MHC (70%) | $73.0 \pm 16.6$ | $84.2 \pm 2.2$ |
|  | ✓ | ER | LWN in MLP | $74.4 \pm 12.2$ | $82.5 \pm 3.0$ |
|  | ✓ | ER | LWN in HCN | $\mathbf{76.6 \pm 13.6}$ | $\mathbf{85.9 \pm 1.6}$ |
|  | ✓ | ER | LWN in MHC (70%) | $76.0 \pm 14.3$ | $85.9 \pm 1.9$ |
| Perm-MNIST | - - | HAT | MLP | $67.2 \pm 20.6$ | $72.3 \pm 18.5$ |
|  | - - | HAT | HCN | $71.6 \pm 15.1$ | $79.2 \pm 6.0$ |
|  | ✓ | HAT | MHC (70%) | $70.5 \pm 15.7$ | $78.8 \pm 6.6$ |
|  | ✓ | HAT | LWN in MLP | $71.5 \pm 17.0$ | $77.6 \pm 12.3$ |
|  | ✓ | HAT | LWN in HCN | $\mathbf{75.7 \pm 13.2}$ | $\mathbf{83.4 \pm 3.8}$ |
|  | ✓ | HAT | LWN in MHC (70%) | $75.3 \pm 13.5$ | $83.2 \pm 4.0$ |

Table 7: More Results with Continual Learning Techniques Applied

| Task | Ours | CL Tech. | Classifier | ACC (%) | FA1 (%) |
|---|---|---|---|---|---|
| Inc-Cifar100 | - - | EWC | MLP | $65.5 \pm 33.4$ | $62.3 \pm 13.8$ |
| (inc. 5 classes) | - - | EWC | HCN | $68.0 \pm 27.9$ | $66.3 \pm 9.7$ |
|  | ✓ | EWC | MHC (10%) | $68.0 \pm 30.0$ | $69.1 \pm 5.0$ |
|  | ✓ | EWC | LWN in MLP | $68.9 \pm 27.7$ | $62.8 \pm 6.9$ |
|  | ✓ | EWC | LWN in HCN | $69.0 \pm 28.4$ | $67.0 \pm 6.4$ |
|  | ✓ | EWC | LWN in MHC (10%) | $\mathbf{69.8 \pm 26.1}$ | $\mathbf{69.9 \pm 5.7}$ |
| Inc-Cifar100 | - - | ER | MLP | $69.5 \pm 23.2$ | $67.1 \pm 6.4$ |
| (inc. 5 classes) | - - | ER | HCN | $70.0 \pm 23.7$ | $67.2 \pm 6.5$ |
|  | ✓ | ER | MHC (10%) | $70.0 \pm 23.4$ | $68.0 \pm 6.4$ |
|  | ✓ | ER | LWN in MLP | $71.0 \pm 23.2$ | $67.5 \pm 4.8$ |
|  | ✓ | ER | LWN in HCN | $70.8 \pm 23.0$ | $67.0 \pm 3.4$ |
|  | ✓ | ER | LWN in MHC (10%) | $\mathbf{71.1 \pm 23.4}$ | $\mathbf{67.7 \pm 6.2}$ |
| Inc-Cifar100 | - - | HAT | MLP | $65.8 \pm 30.3$ | $60.4 \pm 12.3$ |
| (inc. 5 classes) | - - | HAT | HCN | $65.9 \pm 33.2$ | $63.3 \pm 7.7$ |
|  | ✓ | HAT | MHC (10%) | $\mathbf{68.3 \pm 27.9}$ | $\mathbf{66.8 \pm 8.3}$ |
|  | ✓ | HAT | LWN in MLP | $66.6 \pm 31.6$ | $61.5 \pm 9.3$ |
|  | ✓ | HAT | LWN in HCN | $66.3 \pm 34.2$ | $65.4 \pm 5.3$ |
|  | ✓ | HAT | LWN in MHC (10%) | $67.3 \pm 30.9$ | $64.9 \pm 7.9$ |

Table 8: More Results with Continual Learning Techniques Applied

| Task | Ours | CL Tech. | Classifier | ACC (%) | FA1 (%) |
|------|------|----------|-----------|---------|---------|
| Inc-Cifar100 | - - | EWC | MLP | $52.1 \pm 17.1$ | $48.7 \pm 11.3$ |
| (inc. 10 classes) | - - | EWC | HCN | $55.4 \pm 14.2$ | $51.0 \pm 8.3$ |
| | ✓ | EWC | MHC (10%) | $56.2 \pm 13.1$ | $51.3 \pm 9.8$ |
| | ✓ | EWC | LWN in MLP | $55.4 \pm 15.7$ | $51.5 \pm 7.1$ |
| | ✓ | EWC | LWN in HCN | $57.3 \pm 13.4$ | $53.5 \pm 8.9$ |
| | ✓ | EWC | LWN in MHC (10%) | $\mathbf{57.9 \pm 12.3}$ | $\mathbf{57.1 \pm 7.4}$ |
| Inc-Cifar100 | - - | ER | MLP | $51.9 \pm 13.8$ | $52.6 \pm 5.9$ |
| (inc. 10 classes) | - - | ER | HCN | $52.9 \pm 13.4$ | $53.2 \pm 8.2$ |
| | ✓ | ER | MHC (10%) | $53.3 \pm 13.2$ | $53.9 \pm 5.8$ |
| | ✓ | ER | LWN in MLP | $54.7 \pm 13.5$ | $\mathbf{54.9 \pm 4.6}$ |
| | ✓ | ER | LWN in HCN | $55.4 \pm 12.8$ | $54.4 \pm 7.3$ |
| | ✓ | ER | LWN in MHC (10%) | $\mathbf{55.6 \pm 12.7}$ | $54.3 \pm 4.1$ |
| Inc-Cifar100 | - - | HAT | MLP | $52.7 \pm 17.3$ | $43.5 \pm 13.6$ |
| (inc. 10 classes) | - - | HAT | HCN | $55.3 \pm 14.5$ | $46.1 \pm 11.8$ |
| | ✓ | HAT | MHC (10%) | $55.8 \pm 13.4$ | $\mathbf{52.0 \pm 12.7}$ |
| | ✓ | HAT | LWN in MLP | $54.1 \pm 17.2$ | $45.9 \pm 8.7$ |
| | ✓ | HAT | LWN in HCN | $\mathbf{56.0 \pm 14.9}$ | $50.8 \pm 10.0$ |
| | ✓ | HAT | LWN in MHC (10%) | $55.8 \pm 14.6$ | $50.9 \pm 12.4$ |

Table 9: More Results with Continual Learning Techniques Applied

| Task | Ours | CL Tech. | Classifier | ACC (%) | FA1 (%) |
|------|------|----------|-----------|---------|---------|
| Inc-Cifar100 | - - | EWC | MLP | $43.3 \pm 15.2$ | $36.1 \pm 5.5$ |
| (inc. 20 classes) | - - | EWC | HCN | $47.0 \pm 10.6$ | $44.0 \pm 4.9$ |
| | ✓ | EWC | MHC (10%) | $47.5 \pm 9.0$ | $44.3 \pm 5.6$ |
| | ✓ | EWC | LWN in MLP | $46.4 \pm 13.9$ | $40.4 \pm 6.9$ |
| | ✓ | EWC | LWN in HCN | $48.8 \pm 9.3$ | $45.2 \pm 2.9$ |
| | ✓ | EWC | LWN in MHC (10%) | $\mathbf{49.2 \pm 8.6}$ | $\mathbf{46.9 \pm 3.2}$ |
| Inc-Cifar100 | - - | ER | MLP | $41.7 \pm 17.0$ | $37.1 \pm 5.5$ |
| (inc. 20 classes) | - - | ER | HCN | $42.1 \pm 16.2$ | $38.5 \pm 5.2$ |
| | ✓ | ER | MHC (10%) | $42.4 \pm 15.5$ | $39.0 \pm 4.0$ |
| | ✓ | ER | LWN in MLP | $43.8 \pm 15.6$ | $39.4 \pm 6.0$ |
| | ✓ | ER | LWN in HCN | $44.9 \pm 15.1$ | $40.7 \pm 4.3$ |
| | ✓ | ER | LWN in MHC (10%) | $\mathbf{45.1 \pm 14.4}$ | $\mathbf{41.6 \pm 4.8}$ |
| Inc-Cifar100 | - - | HAT | MLP | $43.4 \pm 13.0$ | $38.0 \pm 6.0$ |
| (inc. 20 classes) | - - | HAT | HCN | $44.2 \pm 13.1$ | $38.6 \pm 8.3$ |
| | ✓ | HAT | MHC (10%) | $45.6 \pm 9.9$ | $43.1 \pm 5.3$ |
| | ✓ | HAT | LWN in MLP | $45.7 \pm 11.8$ | $41.5 \pm 5.8$ |
| | ✓ | HAT | LWN in HCN | $47.0 \pm 10.5$ | $43.5 \pm 7.6$ |
| | ✓ | HAT | LWN in MHC (10%) | $\mathbf{47.3 \pm 9.1}$ | $\mathbf{44.9 \pm 4.8}$ |

Table 10: More Results with Continual Learning Techniques Applied

| Task | Ours | CL Tech. | Classifier | ACC (%) | FA1 (%) |
|---|---|---|---|---|---|
| Inc-CUB200 | - - | EWC | MLP | $57.7 \pm 46.3$ | $43.3 \pm 25.1$ |
| (inc. 10 classes) | - - | EWC | HCN | $63.0 \pm 42.2$ | $49.8 \pm 14.1$ |
|  | ✓ | EWC | MHC (10%) | $63.2 \pm 40.9$ | $57.6 \pm 19.7$ |
|  | ✓ | EWC | LWN in MLP | $73.6 \pm 37.5$ | $70.2 \pm 14.4$ |
|  | ✓ | EWC | LWN in HCN | $\mathbf{77.4 \pm 31.5}$ | $76.3 \pm 13.3$ |
|  | ✓ | EWC | LWN in MHC (10%) | $76.9 \pm 31.8$ | $\mathbf{77.0 \pm 12.2}$ |
| Inc-CUB200 | - - | ER | MLP | $71.3 \pm 36.2$ | $76.3 \pm 9.1$ |
| (inc. 10 classes) | - - | ER | HCN | $71.7 \pm 35.5$ | $78.5 \pm 7.5$ |
|  | ✓ | ER | MHC (10%) | $72.7 \pm 36.0$ | $79.6 \pm 6.4$ |
|  | ✓ | ER | LWN in MLP | $76.1 \pm 32.1$ | $78.2 \pm 5.7$ |
|  | ✓ | ER | LWN in HCN | $\mathbf{78.2 \pm 30.3}$ | $82.5 \pm 6.5$ |
|  | ✓ | ER | LWN in MHC (10%) | $77.9 \pm 30.7$ | $\mathbf{82.6 \pm 6.4}$ |
| Inc-CUB200 | - - | HAT | MLP | $60.9 \pm 43.0$ | $48.8 \pm 25.8$ |
| (inc. 10 classes) | - - | HAT | HCN | $62.3 \pm 42.1$ | $55.8 \pm 18.5$ |
|  | ✓ | HAT | MHC (10%) | $66.5 \pm 39.0$ | $66.1 \pm 13.5$ |
|  | ✓ | HAT | LWN in MLP | $\mathbf{75.2 \pm 33.8}$ | $\mathbf{82.2 \pm 17.8}$ |
|  | ✓ | HAT | LWN in HCN | $72.0 \pm 40.9$ | $81.4 \pm 12.9$ |
|  | ✓ | HAT | LWN in MHC (10%) | $74.0 \pm 36.1$ | $81.2 \pm 9.4$ |

Table 11: More Results with Continual Learning Techniques Applied

| Task | Ours | CL Tech. | Classifier | ACC (%) | FA1 (%) |
|---|---|---|---|---|---|
| Inc-CUB200 | - - | EWC | MLP | $57.8 \pm 24.2$ | $53.4 \pm 14.7$ |
| (inc. 20 classes) | - - | EWC | HCN | $61.1 \pm 23.3$ | $55.7 \pm 12.4$ |
|  | ✓ | EWC | MHC (10%) | $62.8 \pm 23.7$ | $60.3 \pm 14.2$ |
|  | ✓ | EWC | LWN in MLP | $74.0 \pm 18.2$ | $69.5 \pm 8.0$ |
|  | ✓ | EWC | LWN in HCN | $\mathbf{76.7 \pm 17.1}$ | $\mathbf{73.0 \pm 10.3}$ |
|  | ✓ | EWC | LWN in MHC (10%) | $76.3 \pm 17.0$ | $72.7 \pm 10.0$ |
| Inc-CUB200 | - - | ER | MLP | $60.3 \pm 24.4$ | $64.9 \pm 8.8$ |
| (inc. 20 classes) | - - | ER | HCN | $62.2 \pm 23.7$ | $67.8 \pm 7.5$ |
|  | ✓ | ER | MHC (10%) | $63.0 \pm 24.3$ | $68.4 \pm 6.1$ |
|  | ✓ | ER | LWN in MLP | $67.5 \pm 23.2$ | $70.4 \pm 6.2$ |
|  | ✓ | ER | LWN in HCN | $72.0 \pm 21.1$ | $74.5 \pm 6.8$ |
|  | ✓ | ER | LWN in MHC (10%) | $\mathbf{72.7 \pm 20.1}$ | $\mathbf{75.0 \pm 8.1}$ |
| Inc-CUB200 | - - | HAT | MLP | $58.6 \pm 24.5$ | $65.4 \pm 10.8$ |
| (inc. 20 classes) | - - | HAT | HCN | $58.1 \pm 26.7$ | $65.3 \pm 13.5$ |
|  | ✓ | HAT | MHC (10%) | $61.3 \pm 24.3$ | $71.4 \pm 7.3$ |
|  | ✓ | HAT | LWN in MLP | $75.0 \pm 22.8$ | $\mathbf{81.1 \pm 7.2}$ |
|  | ✓ | HAT | LWN in HCN | $\mathbf{75.6 \pm 18.7}$ | $77.2 \pm 8.4$ |
|  | ✓ | HAT | LWN in MHC (10%) | $75.5 \pm 18.6$ | $79.0 \pm 10.7$ |

Table 12: More Results with Continual Learning Techniques Applied

| Task | Ours | CL Tech. | Classifier | ACC (%) | FA1 (%) |
|---|---|---|---|---|---|
| Inc-CUB200 | - - | EWC | MLP | $56.0 \pm 17.8$ | $46.8 \pm 13.5$ |
| (inc. 40 classes) | - - | EWC | HCN | $58.8 \pm 13.7$ | $51.8 \pm 6.6$ |
| | ✓ | EWC | MHC (10%) | $62.3 \pm 11.0$ | $57.4 \pm 4.9$ |
| | ✓ | EWC | LWN in MLP | $72.7 \pm 17.7$ | $62.4 \pm 8.8$ |
| | ✓ | EWC | LWN in HCN | $76.8 \pm 14.3$ | $68.4 \pm 6.0$ |
| | ✓ | EWC | LWN in MHC (10%) | $\mathbf{77.4 \pm 14.0}$ | $\mathbf{69.7 \pm 7.7}$ |
| Inc-CUB200 | - - | ER | MLP | $51.9 \pm 20.1$ | $47.6 \pm 4.5$ |
| (inc. 40 classes) | - - | ER | HCN | $52.0 \pm 20.2$ | $48.2 \pm 9.0$ |
| | ✓ | ER | MHC (10%) | $53.4 \pm 20.5$ | $49.6 \pm 8.4$ |
| | ✓ | ER | LWN in MLP | $64.7 \pm 25.3$ | $53.9 \pm 7.7$ |
| | ✓ | ER | LWN in HCN | $70.0 \pm 21.4$ | $60.6 \pm 8.1$ |
| | ✓ | ER | LWN in MHC (10%) | $\mathbf{71.8 \pm 20.0}$ | $\mathbf{63.1 \pm 8.6}$ |
| Inc-CUB200 | - - | HAT | MLP | $55.7 \pm 16.1$ | $59.6 \pm 6.1$ |
| (inc. 40 classes) | - - | HAT | HCN | $56.4 \pm 14.1$ | $62.1 \pm 7.4$ |
| | ✓ | HAT | MHC (10%) | $58.8 \pm 15.0$ | $65.5 \pm 6.3$ |
| | ✓ | HAT | LWN in MLP | $75.9 \pm 11.8$ | $\mathbf{79.1 \pm 6.0}$ |
| | ✓ | HAT | LWN in HCN | $77.6 \pm 9.5$ | $76.1 \pm 5.6$ |
| | ✓ | HAT | LWN in MHC (10%) | $\mathbf{78.8 \pm 7.7}$ | $77.5 \pm 4.6$ |

## E   THE NOISE INJECTION SETUP

The NI setup of Kuo et al. (2021a) can be seen as a direct extension of the Perm-MNIST task. Based on the description of that paper, there is only one extra step – which is to randomly shuffle a portion of the content of the Perm-MNIST vectors. To elaborate, each task of Perm-MNIST first reorganises each MNIST image as a vector of 784 pixels; it then applies the permutation mask of the current task to that vector of pixels. On top of this, NI randomly selects a portion of the pre-processed 784 pixels and shuffle those selected pixels.

The remaining hyperparameters of the NI scenario were identical to our normal Perm-MNIST setup. There were 20 tasks in total. We naïvely sequentially trained the classifiers over 5K images on the first task, then observe 1K images on the rest of the tasks. All classifiers had two layers with hidden dimension 100; and they were updated with SGD with learning rate 0.01.

## F   ADDITIONAL EXPERIMENTAL SETUPS

Table 13: More Results with Continual Learning Techniques Applied

| Task | Ours | CL Tech. | Classifier | ACC (%) | FA1 (%) |
|---|---|---|---|---|---|
| Perm-MNIST | - - | EBM | - - | $74.2 \pm 12.2$ | $61.1 \pm 13.7$ |
| | - - | EBM | HCN | $77.7 \pm 10.0$ | $65.8 \pm 11.1$ |
| | ✓ | EBM | MHC (70%) | $77.9 \pm 9.9$ | $66.8 \pm 14.4$ |
| | ✓ | EBM | with LWN | $80.3 \pm 6.3$ | $74.9 \pm 6.1$ |
| | ✓ | EBM | LWN in HCN | $\mathbf{82.3 \pm 5.2}$ | $\mathbf{77.0 \pm 4.2}$ |
| | ✓ | EBM | LWN in MHC (70%) | $82.1 \pm 5.3$ | $76.6 \pm 3.2$ |
| Inc-Cifar100 | - - | EBM | - - | $31.3 \pm 22.4$ | $21.8 \pm 9.4$ |
| (inc. 10 classes) | ✓ | EBM | LWN in MHC (10%) | $\mathbf{37.4 \pm 19.6}$ | $\mathbf{31.6 \pm 3.2}$ |
| Inc-CUB200 | - - | EBM | - - | $32.4 \pm 22.4$ | $15.15 \pm 4.7$ |
| (inc. 20 classes) | ✓ | EBM | LWN in MHC (10%) | $\mathbf{51.3 \pm 19.5}$ | $\mathbf{30.8 \pm 5.5}$ |

It was suggested by one of our reviewers to include additional experiments to test our techniques on the non-standard continual learning classifiers. To this end, we supplemented our existing experiments with *Energy-Based Models* (EBM) (Li et al., 2020) as alternative classifiers.

As shown in Figure 1, traditional continual learning classifiers used softmax as their final layer. However, it was noted in Li et al. (2020) that softmax classifiers encourage a *winner-takes-all* dy-

namic for network parameterisation. Thus alternatively, they advocated for the use of EBMs. The aim of their EBM was to minimise the energy for the input class while rising the energy level for all remaining classes. Such a setup would allow the network to lower its confidence in making an incorrect decision.

However, their EBM required a paradigmatic shift towards the conventional network setup for continual learning. First, they relinquished softmax as their final layer and instead replaced it with a linear transformation with a single output dimension. Second, they required both the data along with their ground-truth labels as input. Regardless of these changes, our MHC and LWN techniques were still applicable to EBM classifiers. As shown in Table 13, our techniques were able to improve the vanilla EBM setup across three different datasets.

