# OpenReview forum: "Designing Less Forgetful Networks for Continual Learning"
_ICLR.cc/2022/Conference — ICLR 2022 Submitted_

### Official Review · Reviewer_tojU · 2021-10-31

**Correctness:** 3
**Technical Novelty And Significance:** 3
**Empirical Novelty And Significance:** 3
**Recommendation:** 8
**Confidence:** 3

**Main Review:**

I find the general approach of "baking" in the ability to handle continual learning into the network architecture quite an elegant way of solving the catastrophic forgetting problem.  Though the proposed modifications to the HCN architecture are rather simple, I think this paper is worthy of note, because these changes make the network quite more robust for continual learning, and thus present a relatively straight forward, headache free (other than deciding on the MHC threshold) architecture that seems significantly less vulnerable to catastrophic forgetting.

The paper would be stronger if there was a bit more theoretical explanation of how the internal dynamics of the masking and normalisation "select" weights in a way that does not destroy what the network has learned.  As it stands, the explanation is mostly based on intuitive understanding of the dynamics and empirical evidence, and quite general principles, akin to the argument that HCN is good for catastrophic forgetting, because it multiplies the derivative by a fraction.  BTW, I find it a bit counter-intuitive that HCN was designed to "withstand gradient vanishing" and is less vulnerable to catastrophic forgetting due to scaling down of derivatives....which, effectively, is gradient vanishing.  It seems that these two statements contradict each other.

While overall the paper is pretty well written, some parts could be improved for clarity.  Mathematical notation seems cumbersome (why use $L_\gamma$  etc. subscript everywhere to indicate layer index...when just the $\gamma$ subscript will do for index identifying the layer.  Why the neuron's output is labelled with $a$ for MLP and with $k$ HCN-based models?   It's exactly same neurons, isn't it.   Also, HCN could be explained a bit better (in Section 2).  Sure, it's always possible to go and read the references...but given that HCN is the architecture that this paper builds on, it could be explained a bit more thoroughly.

**Summary Of The Paper:**

The paper proposes modifications to the highway connection network (HCN) architecture, which exhibits some inherent resistance to catastrophic forgetting.  These modification introduce fairly simple masking rule for selection of portion of weights to update and normalisation that (presumably) further adjust the internal representation to be more selective in which weights to update for a given task.  Empirical evaluation shows that these modification are highly effective in alleviating catastrophic forgetting.

**Summary Of The Review:**

It's a decent paper - it would be better if there was more theoretical explanation for why the proposed changes to the architecture help with continual learning, but the provided empirical evaluation seems pretty good, and the proposed architectural changes to HCN seems quite effective.

---

> ### Author Response · Authors · 2021-11-17
> **Thread # 1 to Reviewer tojU**
>
> The authors of this manuscript thank Reviewer tojU for your review.
> Please find our replies to your comments below.
>
> It seems like there were 2 main concerns.
>
>    	(1) On the notation of our manuscript.
>      	(2) On the theoretical relations between
>  		MHC, HCN, catastrophic forgetting, and gradient vanishing.
>
> We will reply to your responses in the 2 threads below.

---

> ### Author Response · Authors · 2021-11-17
> **Thread # 2 to Reviewer tojU**
>
> This thread will elaborate on the choice of our notations.
>
> Regarding your comment:
> “Mathematical notation seems cumbersome (why use etc. subscript everywhere to indicate layer index...when just the subscript will do for index identifying the layer.”
>
> The mathematical notations in our manuscript were largely adopted from page 2 in Kuo et al (2021b)’s paper. The authors’ of that paper used
>
>     	a_{L_{\gamma}} to represent
>      	the post-activated neural values at layer \gamma.
>
> This 3-layered notation was adopted to prevent confusion. Consider \omega_i in Equation (1) on page 2 of our manuscript (also adopted from page 2 of Kuo et al).
>
>  	\omega_i represented
>      	the dataset of the i-th task in the continuum.
>
> So, if we were to write a_{L_{\gamma}} as a_\gamma, it would be likely that the readers would mis-interpret
>
>  	a_\gamma as
>      	the \gamma-th post-activated neuron.
>
> This further linked to another comment:
> “Why the neuron's output is labelled with “a” for MLP and with “k” HCN-based models? It's exactly same neurons, isn't it.”
>
> Again, our notation was adopted from page 2 of Kuo et al (2021b). In that paper, it was actually important for Kuo et al to explicitly differentiate between the
>
>   	post-activated neural values for MLP and the
>    	post-activated neural values for HCN.
>
> This was because that the partial derivative of the post-activated neural values with respect to the weights were different. More specifically,
>
>  	the post-activated neural values of HCN were much lower than that of MLP
>  	because it was scaled by the gated unit of HCN.
>
> More details about this argument could be found in the last paragraph of Section 3 on page 3 of our manuscript.

---

> ### Author Response · Authors · 2021-11-17
> **Thread # 3 to Reviewer tojU**
>
> This thread will elaborate on the relation between catastrophic forgetting and gradient vanishing.
>
> Regarding your comment:
> “BTW, I find it a bit counter-intuitive that HCN was designed to "withstand gradient vanishing" and is less vulnerable to catastrophic forgetting due to scaling down of derivatives....which, effectively, is gradient vanishing. It seems that these two statements contradict each other.”
>
> This is actually one of the more theoretical parts of our paper; and that gradient vanishing and catastrophic forgetting can be considered as “two-sides of the same coin” in network optimisation. In order to highlight the differences between gradient vanishing and catastrophic forgetting, we have added underbraces and overbraces towards Equation 3 on page 3 of our manuscript. Please find our extended explanation below.
>
> While HCN was capable of preventing catastrophic forgetting, it was definitely not doing so through encouraging gradient vanishing. As shown in Equation 3 on page 3 of our paper, the effective gradient update was
>
>  	The derivative of C w.r.t. W_{L_\eta} =
>           	(a)     	the derivative of C w.r.t. y_hat, times
>           	(b)  	a product of Jacobians,
>                		the derivative of
>              		new layer output (a_{L_{\eta + 1}}) w.r.t
>              		preceding layer output (a_{L_\eta}), times
>           	(c)  		the derivative of
>              		the output of the target layer (a_{L_\eta}) w.r.t
>              		the weights of the target layer (W_{L_\eta}).
>
> Note, element (b) corresponded to the [.] term in Equation 3 while element (c) correspond to the {.} term in Equation 3.
>
> Previously,
>
>      	residual connections (He et al, 2016) <1> and
>      	highway connections (Srivastava et al, 2015) <2>
>
> stop gradient vanishing because both architectural design introduced the identity matrix to the Jacobian terms in element (b).
>
> However, in Kuo et al (2021b), the authors of that paper showed that HCN could mitigate catastrophic forgetting because the gated unit was introduced in element (c). The contribution to their paper was hence orthogonal to residual connections and highway connections. This was because that
>
>  	residual connections and highway connections prevent gradient vanishing
>      		occurring in a series of products (element (b));
>      	while HCN mitigated forgetting by introducing a “one-off” scale-down
>           	in element (c).
>
> Essentially, what Kuo et al (2021b) had found in their paper was that there was a close connection between gradient vanishing and catastrophic forgetting.
>
>  	If a repetitive gradient scale down was applied, we get gradient vanishing;
>      	but if a one-off gradient scale down was applied, we could mitigate
>  		forgetting.
>
> Based on the findings of Kuo et al, our paper aimed to take advantage of this HCN property and consider the interplay between HCN’s gated unit and the structural relationships between different layers of neural network modules.

---

> > ### Comment · Reviewer_tojU · 2021-11-26
> > **Thanks for the response**
> >
> > The authors clarified the relationship between vanishing gradient and catastrophic forgetting in their equations.  I think it's a good paper and raise my score.

---

### Official Review · Reviewer_pQdx · 2021-11-02

**Correctness:** 2
**Technical Novelty And Significance:** 3
**Empirical Novelty And Significance:** 2
**Recommendation:** 6
**Confidence:** 4

**Details Of Ethics Concerns:**

1. What is \omeg_i in definition (1). And wherein the method we use such information?
2. Why activation is denoted by \H and natural number are marked by \gamma instead of classical n,k, l.
3. It is not natural to use three-level indexes A_{L_{\Gamm}}
4. I do not underfund notation in equation (3). The note is nonintuitive for me.
5. Masked Highway Connection seems to be a simple modification of HCN. However, it also requires adding new parameters to neural networks.
6. Also, Layer-Wise Normalisation seems to be classical normalization. Why does it work in CL setting?
7. The authors use a very small neural network for EWC and other classical approaches in the experimental section.  Also, we use only a dense layer (as a trainable layer) without convolution ones.
8. Authors use only SGD. Why not ADAM?
9. It is not obvious that all neural networks in the experimental section have similar parameters.
10. Authors tested MHCs with a range of different settings. Hot the parameter shud be tune in practice?

**Main Review:**

The paper is hard to read and uses very strange notation. It is very hard to understand formulas since authors do not use standard notation in deep learning. The modification proposed in the paper seems small and can be understood as an effect of architecture search.

The most interesting part is the experimental section. The authors have verified six classifier networks with four continual learning setups over three datasets. Unfortunately, it is not clear why the authors use SGD instead of adam. It is unclear why we do not have results on a classical convolutional neural network with 5-10 layers. It is not clear why in  MHC, authors train models with additional parameters. It is not clear the influence of such parameter since in Tab. 1 on Perm-MNIST wins LWN in MHC (70%) and on Inc-Cifar100 and Inc-CUB200 LWN in MHC (10%). It is better to choose a small or large percent of gated unit activation. Hot the parameter shud be tune in practice?

**Summary Of The Paper:**

In the paper, the authors present two modifications: Masked Highway Connection and Layer-Wise Normalisation. In Masked Highway Connection, authors add a binary mask to classical HCN (Highway Connection Networks) and slightly change the training procedure. On the other hand, Layer-Wise Normalisation is a new normalization of activation in the neural network.

**Summary Of The Review:**

The paper is hard to read and uses strange notation. The proposed modification seems to be small and not significant. The experimental section looks nice and is convincing, but authors should use some additional architecture and show how the method works on adam optimizer.

---

> ### Author Response · Authors · 2021-11-17
> **Thread # 1 to Reviewer pQdx**
>
> The authors of this manuscript thank Reviewer pQdx for your review.
> Please find our replies to your comments below.
>
> It seems like there were 3 main concerns.
>
>  	(1) On the notation of our manuscript.
>      	(2) On the experimental settings of our paper.
>      	(3) On the best practices and utilities of MHC and LWN.
>
> We will provide some short feedback below,
> and we will elaborate our explanations over the next three threads.
>
> Please note, your concerns were raised in “Details of Ethics Concern”. However, we think that the specific points that you have made under this discrimination, bias, or fairness category are more about technicalities than ethics. In addition, none of the other 4 reviewers have reported any ethics concern.
>
> However, we address these specific points as follows:
>
>  	Regard point (1),
>  		we adopted the notations therein Kuo et al (2021b)’s HCN paper
>  		because the new methods introduced in our paper was directly
>  		extended from HCN.
>  		This was acknowledged by Reviewer zxFC with
>  		“...In my opinion the notation is Section 3 is a bit confusing, but I think
>  		the points made in Kuo et al. (2021b) were well summarized (so using  		the notation from that paper is justified)....”
>
>  	Regarding point (2),
>  		the experimental setup of our paper followed that in Kuo et al (2021b).
>  		Specifically, Kuo et al’s paper followed the network
>  		compartmentalisation experimental setup of Mai et al (2020)
>  		(commented in paragraph 1 in page 2 of our manuscript).
>  		Mai et al’s network compartmentalisation is extremely powerful; and
>  		this setup has previously helped them to win the
>  		CLVision CVPR Workshop challenge in 2020
>  		(see
>  	https://sites.google.com/view/clvision2020/challenge/challenge-winners)
>
>  	Regarding point (3),
>  		we will generally advise other practitioners to use the version of MHC  		that allows only 10% modification in gradient descent. This is because  		that in Table 1 on page 7, we showed that all MHC performances were  		either on par or better than HCN.
>
> We will further elaborate the technicalities of your concerns in details over the next three threads.
>
> Your concerns 1., 2., 3., 4., will be addressed in Thread 2;
>
> your concerns 5, 7., 8., 9., will be addressed in Thread 3; and
>
> your concerns 6., 10., will be addressed in Thread 4.

---

> > ### Comment · Reviewer_pQdx · 2021-11-24
> > **Thank you for your response.**
> >
> > The authors answered all my questions. Moreover, the authors clarify some misleading information. After introducing the above corrections, the paper will be more precise for me. Therefore, I rise my score.

---

> ### Author Response · Authors · 2021-11-17
> **Thread # 2 to Reviewer pQdx**
>
> This thread will provide more details on the notations of point (1) in Thread 1.
>
> Regarding your comment:
> “1. What is \omeg_i in definition (1). And wherein the method we use such information?”
>
> The notation \omega_i was introduced in Equation (1) in Section 2 on page 2 of our manuscript. Please note that \omega_i is not a hyper-parameter to any of the methods. As we stated in Section 2,
>
>    	\omega_i referred to the
>      	dataset of the i-th task in the
>
> continuum of tasks for the continual learning setup.
>
> Regarding your comment:
> “The paper is hard to read and uses very strange notation”
> This was previously answered in our brief answer in Thread 1.
> All of our notation follows the original work of Kuo et al (2021b); more
> specifically, refer to Section 4.1 on page 2 in their paper.
>
> This further links to your comments:
>
>  	2.     Why activation is denoted by \H and natural number are marked by
>          		\gamma instead of classical n,k, l.
>      and
>      	3.      It is not natural to use three-level indexes A_{L_{\Gamm}}
>
> of which are both related to Equation 6 on page 2 of Kuo et al (2021b).
>
> According to our observation, Kuo et al (2021b) used the lower case \gamma for a natural number because “n” and “k” were denoted as the post-activation neural values for different backbone network architectural designs.
>
> \H was used in their paper because this was also used in the original Highway Network paper by Srivastava, Greff, and Schmidhuber (2015) <1>;
>
> refer to Equation (1) on page 2 of the 2015 Highway Network paper.
>
> Furthermore, Kuo et al’s three-level indexes of A_{L_{\Gamm}} should be interpreted as
>
> the post-activated neural values in layer \Gamma (L_\Gamma)
>
> <1>:     Srivastava, Greff, and Schmidhuber.     Highway Networks.
>
> <<<>>>
>
> Regarding your comment:
> “4. I do not underfund notation in equation (3). The note is nonintuitive for me.”
>
> Our Equation (3) on page 3 of our manuscript is the same the equation in
> Section 4.1 on page 2 of Kuo et al (2021b).
>
> This is actually a relatively well-known equation known originally for describing the gradient vanishing problem. This equation can be found in
>
> the highly cited paper by Pascanu et al (2013) <2> -- see Equation (4) on page 2 of that paper.
>
> This equation also appeared in the highly cited LSTM paper (Hochreiter & Schmidhuber, 1997) <3> -- first appearing in Equation (24) in page 27 of that paper; then reappearing in Equation (32) in page 28, Equations (38) and (41) in page 29 of that paper.
>
>  	<2>: Pascanu et al.
>  		On the Difficulty of Training Recurrent Neural Networks.
>  	<3>: Hochreiter & Schmidhuber.
>  		Long Short-Term Memory.

---

> ### Author Response · Authors · 2021-11-17
> **Thread # 3 to Reviewer pQdx**
>
> This thread will provide more details on the experimental setups of point (2) in Thread 1.
>
> Regarding your comments:
> “It is unclear why we do not have results on a classical convolutional neural network with 5-10 layers” and
> “7. The authors use a very small neural network for EWC and other classical approaches in the experimental section. Also, we use only a dense layer (as a trainable layer) without convolution ones.”
>
> As mentioned in the short answers in Thread 1, the experimental setup in our manuscript followed the setup of Kuo et al (2021b). In addition, Kuo et al’s paper
> followed Mai et al (2020) (cited in paragraph 1 in page 2 of our manuscript); and
> that Mai et al’s network compartmentalisation previously helped them won the CLVision CVPR Workshop challenge in 2020.
>
> Furthermore, in page 4 of Mai et al’s paper, the authors’ of that paper found that better results could be achieved when the shallower layers (the pre-trained feature extractor) was not fine-tuned. Their experiment was conducted over multiple continual learning scenarios and the results in Tables 1, 2, 3 in page 4 of their paper showed that, regardless of the continual learning scenario, network compartmentalisation always achieved much better results.
>
> <<<>>>
>
> Regarding your comments:
> “Unfortunately, it is not clear why the authors use SGD instead of adam.”
> and
> “8. Authors use only SGD. Why not ADAM?”
>
> It is actually a common practice in the field of continual learning to use SGD over ADAM.
> For instance, in the well-cited GEM paper by Lopez-Paz and Ranzato (2017)
> All of their methods on their official repository were implemented with SGD.
>
> See Lines 178-179 in their code
>
> https://github.com/facebookresearch/GradientEpisodicMemory/blob/master/main.py
>
> To elaborate,
> they implemented SGD to train 3 continual learning techniques, including GEM and also another two well-known methods of EWC and iCaRL.
>
> In the official repository of Chaudry et al (2018), the authors also
> defaulted their optimiser as SGD for 2 different methods.
> See Line 40 in
>
> https://github.com/facebookresearch/agem/blob/main/fc_permute_mnist.py
>
> and see Line 41 in
>
> https://github.com/facebookresearch/agem/blob/main/conv_split_cifar.py
>
> Yet in another instance, the official repository of HAT by Serra et al (2018) also used SGD in their implementation.
> See Line 38 - 40 in
>
> https://github.com/joansj/hat/blob/master/src/approaches/hat.py
>
> <<<>>>
>
> Regarding your comment:
> “9. It is not obvious that all neural networks in the experimental section have similar parameters.”
> “5. Masked Highway Connection seems to be a simple modification of HCN. However, it also requires adding new parameters to neural networks.”
>
> Please note that our MHC method does not introduce any new parameters to HCN.
> As documented in paragraph 2 in page 4 of our paper, the masking mechanism of MHC was based on the top-K function. The top-K function did not require any new parameters; our MHC network thus had the identical amount of parameters to Kuo et al (2021b)’s HCN.

---

> ### Author Response · Authors · 2021-11-17
> **Thread # 4 to Reviewer pQdx**
>
> This thread will provide more details on the experimental setups of point (3) in Thread 1.
>
> Regarding your comments:
> “It is not clear the influence of such parameter since in Tab. 1 on Perm-MNIST wins LWN in MHC (70%) and on Inc-Cifar100 and Inc-CUB200 LWN in MHC (10%).”
>
> Please note that two points should be simultaneously considered while reading Table 1 on page 7 of our manuscript.
>
>  	First, while our MHC always performed either on-par or better than the
>  			baseline HCN, MHC was also proposed to explore sparsity for
>  			continual learning.
>  	Second, the results in Table 1 thus show that a backbone network could
>  			always achieve either on-par or better performances than when it
>  			only received minimalistic updates under the continual learning
>  			scenario.
>
> This links to another comment:
> “10. Authors tested MHCs with a range of different settings. How the parameter should be tuned in practice?"
>
> In general, we would advise a practitioner to enable only 10% weight update with MHC during gradient descent. Though MHC required 70% update to reach its peak performance for Perm-MNIST, this was more likely to do with the simplicity for the MNIST dataset. As shown in Tables 4 - 12 in the appendices of our manuscript, it was clear that for the contextually rich Cifar100 and CUB200 datasets, it was the lower the % the better the accuracy.
>
> <<<>>>
>
> Regarding your comment:
> “6. Also, Layer-Wise Normalisation seems to be classical normalization. Why does it work in CL setting?”
>
> The reason why LWN worked well for continual learning was elaborated in Section 5 in pages 4 - 5 of our manuscript. Specifically, LWN worked well because the features were “redistributed” across all feature dimensions thus smoothing out all of the potential differences between tasks.
>
> However, it should be noted that not all classical normalisation were good for continual learning. In our paper, we highlighted the danger of using BN on page 8. That is, LWN was specifically required and it shouldn’t be naively replaced by another normalisation technique.

---

### Official Review · Reviewer_zxFC · 2021-11-02

**Correctness:** 4
**Technical Novelty And Significance:** 4
**Empirical Novelty And Significance:** 4
**Recommendation:** 8
**Confidence:** 3

**Main Review:**

This paper excels at making clear the importance of each of the proposed changes to HCNs. The experimentation is thorough, showing strong performance of the proposed methods on several datasets. The motivation for masking and layer normalization is made clear and is supported by empirical measurements.

Very minor weaknesses include small grammatical errors and tense confusion. (Examples listed below).

In general, this paper makes a compelling case MHC and LWN. The experiments span several datasets as well as different external continual learning techniques. The improvements proposed show strong performance in every case considered. In my opinion the notation is Section 3 is a bit confusing, but I think the points made in Kuo et al. (2021b) were well summarized (so using the notation from that paper is justified). Figures 3 and 4 are a bit hard to read, and perhaps representative quantities could be made explicitly clear in the text rather than only using phrases like "deeper colours" (and less critically I think the fonts should be larger in those plots).

These minor points do not affect my score.

(i) In Section 1, the past and present tenses are both used to describe things done in this paper -- this should be consistent.

(ii) Section 1 has a sentence that starts "Our study focuses on designing new architectures in the forward pass which indirectly but positively affects..." I think the conjugation should be 'new architectures... affect'.

(ii) In the beginning of Section 4, it says "Overleaf in the top half of Figure 2... " and this may be a typo.

(Note that I am not an expert in continual learning and none of my own research has been in this space. With that in mind, I admit that I am not up to date or extremely familiar with existing related work.)

**Summary Of The Paper:**

This paper proposed two architectural improvements for networks designed for continual learning. Specifically, binary masks and layer normalization are added to highway connection classifier networks in order to prevent forgetting. The value of each of these elements is made clear from thorough experimentation.

**Summary Of The Review:**

I think this is a strong paper with compelling experimental results. I vote to accept it. Some small grammatical errors should be corrected.

---

> ### Author Response · Authors · 2021-11-17
> **Thread # 1 to Reviewer zxFC**
>
> The authors of this manuscript thank Reviewer zxFC for your review.
> Please find our replies to your comments below.
>
> First, we have made adjustments to our grammatical errors; and thank you for listing the locations of the errors in the manuscript.
>
> Another concern seems to be the descriptions of Figures 3 and 4. More specifically, you mentioned that you would like us to include some representative quantities to describe the figures. We have thus made the following modifications
>
>  	(1) The x-axes of Figures 3 and 4 were modified to improve readability.
>  	(2) We made modifications to paragraph 2 in page 5 of our manuscript;
>  		including
>       	“Each row generally became deeper and this showed that ICS occurred during
>  		re-parameterisation and that the learnt mapping for MNIST images B_{MNIST} had diverged.
>  		After fine-tuning, we found that the divergence was Diff_{act} = 86.44 in MLP and
>  		Diff_{act} = 69.03 in LWN. Forgetting was hence 20%(= [1 - 69:03=86:44] x 100%) less
>  		severe in LWN than in MLP.”
>  	(3) Similar to point (2), we modified the text in paragraph 2 in page 8 to  		include
>        	“BN in MLP had much deeper colours than LWN in MLP. In addition, the divergence was
>  		Diff_{act}= 97.66 in BN in MLP and forgetting was hence
>  		41%(= [1 - 69:03=97:66] x 100%) more severe than LWN in MLP.”
>  	(4) In order to make space for the changes in point (2),
>      	 	we have decided to move the original Figure 3(c) in the old manuscript
>  		and its supporting text to Appendix B on page 13 of the revised
>  		manuscript.

---

> > ### Comment · Reviewer_zxFC · 2021-11-21
> > **Thank you for the response**
> >
> > Thanks for the thorough response! I will maintain my score.

---

### Official Review · Reviewer_h7Ku · 2021-11-04

**Correctness:** 3
**Technical Novelty And Significance:** 2
**Empirical Novelty And Significance:** 3
**Recommendation:** 5
**Confidence:** 4

**Main Review:**

The findings in the paper are useful, adding evidences to the existing literature on continual learning that we need to pay attention to the interplay between learning algorithms and backbones. The introduction of the mask on the gating function in Highway Networks is interesting. However, this seems to be specific to this particular architecture. Also I would like to see more theoretical analysis of the empirical findings.


**Summary Of The Paper:**

The paper investigates the backbone networks that are less prone to catastrophic forgetting. Two modifications to existing backbones are found to be useful: a mask attached to the gate function of the Highway Network, and layer-norm (without tuning parameters). Experiments on top of existing popular learning algorithms designed for continual learning (EWC, ER and HAT) show that these modifications work.

**Summary Of The Review:**

The findings are interesting, suggesting that research on continual learning should pay more attention to designing neural architectures that are natively suitable to handle catastrophic forgetting. However, the novelty seems to be limited as it is based on well-known architectures and ideas.

---

> ### Author Response · Authors · 2021-11-17
> **Thread # 1 to Reviewer h7Ku**
>
> The authors of this manuscript thank Reviewer h7Ku for your review.
> Please find our replies to your comments below.
>
> The main concern seems to be centred around the level of novelty of our manuscript. It is true that our paper is centred around two existing modelling techniques -- namely highway connection and layer normalisation. However, it should be noted that though these two techniques have existed for a while, their full utility and applications remain largely unknown in the field of continual learning.
>
> In order to clarify our research objectives of the paper, we have revised our wording in paragraphs 2 and 3 in page 2 of our manuscript.
> More specifically, we now include
> " We propose the Masked Highway Connection (MHC) and Layer-Wise Normalisation (LWN)
> shown in Figures 1(c) and 1(d) respectively. While similar techniques exist for general machine
> learning; their applications for continual learning are not well studied."
>
> <<<>>>
>
> While our paper remains largely experimental, we provide additional details below on the significance of our findings and how our findings serve as nice supplements to existing theoretical work.
>
> To the best of our knowledge, Kuo et al (2021b) was the first paper that highlighted the unique and inherent advantages of highway connections networks (HCNs) for continual learning. Though that paper demonstrated some convincing results with HCNs, their manuscript was a short paper and the best practices as well as
> limitations for the HCN architecture were not previously explored.
>
> One part of our contribution was about attempting to explain the effectiveness of their method through establishing a connection with the influential Lottery Ticket Hypothesis (LTH) of Frankle & Carbin (2019) <1> (see our discussion in the Related Work Section on page 9 of our manuscript). By masking and limiting the available changes that can be made during gradient descent, our experiments found that continual learning could be made possible even when only 10% of modifications were allowed in gradient descent. We consider this discovery to be relatively significant; because it implied that most features heuristically learnt by the backbone network were transferable while only a very few non-transferrable features were the source cause of catastrophic forgetting.
>
> Another important discovery found in our experiments was on the selection of appropriate normalisation techniques. As shown in Table 3 in page 8 of our manuscript, we demonstrated that the highly popular and influential batch normalisation (BN) (Ioffe & Szegedy, 2015) was unfit for learning multiple tasks in a sequential fashion. In addition, we highlighted that
> normalisation techniques could still be beneficial for the continual learning setup;
> however, practitioners should select layer normalisation instead of BN.
>
>
> <1>: Frankle & Carbin.     The Lottery Ticket Hypothesis: Finding Sparse,
>                  Trainable Neural Networks
> <2>: Ioffe & Szegedy.      Batch Normalisation: Accelerating Deep Network
>                  Training by Reducing Internal Covariate Shift

---

### Official Review · Reviewer_Homk · 2021-11-09

**Correctness:** 3
**Technical Novelty And Significance:** 3
**Empirical Novelty And Significance:** 3
**Recommendation:** 5
**Confidence:** 5

**Main Review:**


* The motivation for the pertained weights and the differences in the experimental setup between datasets is not clear.
* Comparison with relevant work is missing: The idea of sparse, selective update of parameters to mitigate catastrophic forgetting and encourage task separability has been explored previously. For example, [1] with energy-based models, [2] with local synaptic plasticity and neuromodulation, [3] learns relevance maps. The UCL approach was also mentioned in the paper but not compared against. A systematic comparison with this class of approaches will help better assessing the proposed approach in terms of accuracy/computational cost.
* Continual learning approaches are typically evaluated in both task- and class-incremental learning settings, it’s not clear from the experiments about which scenarios is considered.
* Classification accuracy might not be the best metrics to evaluate the continual leaning potential. Other metrics such as the backward and forward transfer are better suited [4].
* Do the results assume a certain task ordering or have they been chosen randomly. Does the accuracy change with the change in relative ordering of the tasks? Will having consecutive having tasks with similar distributional properties have any effect on the interference with previous tasks.


[1] Li, S., et al. "Energy-Based Models for Continual Learning." arXiv preprint arXiv:2011.12216, 2020

[2] Madireddy, S., et al. “Neuromodulated Neural Architectures with Local Error Signals for Memory-Constrained Online Continual Learning”, arXiv preprint arXiv:2007.08159, 2021

[3] Kaushik, Prakhar, et al. "Understanding Catastrophic Forgetting and Remembering in Continual Learning with Optimal Relevance Mapping." arXiv preprint arXiv:2102.11343 (2021).

[4] Mai, Zheda,  et al. "Online Continual Learning in Image Classification: An Empirical Survey." arXiv preprint arXiv:2101.10423 (2021).


**Summary Of The Paper:**

This work proposes modifications to the highway connection networks (HCN) through the introduction of masks for selective weight updates and a layer-wise normalization to mitigate internal covariate shifts at feature level. Experiments were performed on the Permuted-MNIST, incremental CIFAR-100 and incremental CUB200 by incorporating this approach alongside EWC, ER and HAT.

**Summary Of The Review:**

The results look promising, but the novelty of this approach seems limited since the use of learnable masks, gating mechanisms, and layer-wise normalization have been previously proposed in the continual learning context. Moreover, the experiments are limited in terms of comparison with other state-of-the-are continual learning methods, evaluation metrics (such as backward and forward transfer) that characterizes the forgetting behavior.

---

> ### Author Response · Authors · 2021-11-17
> **Thread #1 to Reviewer Homk**
>
> The authors of this manuscript thank Reviewer Homk for your review.
> Please find our replies to your comments below.
> Our replies are broken down in 6 separate threads; some are quick and short, and some and lengthier to discuss more complicated issues.
>
> With regards to your comment:
> The motivation for the pertained weights and the differences in the experimental setup between datasets is not clear.
>
> <<<>>>
>
> We have pre-trained weights in our backbone networks because
> we adopt the network compartmentalisation as advocated in
> Mai et al (2020) (see paragraph 1 on page 2 of our manuscript).
> Network compartmentalisation is an extremely powerful setup; and
> this setup has previously helped Mai et al to win the
> CLVision CVPR Workshop challenge in 2020
> (see https://sites.google.com/view/clvision2020/challenge/challenge-winners).
>
> In addition to network compartmentalisation, Mai et al tried another scenario where they fine-tune the previously fixed weights (see Tables 1, 2, 3 in page 4 of that paper); and they found out that fine-tuning the feature extraction layers usually resulted in less favourable performances.
>
> Another reason why we choose to adopt Mai et al’s network
> compartmentalisation is because that Kuo et al (2021b)’s HCN paper also
> adopted the same practice. Since we consider our methods to be direct
> extensions of Kuo et al’s HCN, we believe that it is necessary to keep the
> same structures for the backbone networks in our own paper as well.
>
> Network compartmentalisation is also generally commonly practiced in continual learning. Below, we discuss 6 published continual learning papers that also adopted this setup.
>
>      In Nguyen et al (2020) <1>,
>           the authors of that paper mentioned that they froze or apply a tiny
>         learning rate on certain layers (see Section 3.2 in <1>).
>      In Golkar et al (2019) <2>,
>          the authors of that paper froze weights in their fixed capacity backbone
>          network (denoted as the blue node and synapses in Figure 2 in <2>).
>           A part of the motivation in <2> was to address the practice of freezing
>          weights and adding modules seen in previous papers <3><4>.
>      In Rusu et al (2016) <3>,
>          continual learning is made possible by freezing weights learnt for old
>          tasks while adding new modules and only training the new modules for
>          new tasks.
>      The scalability was slightly addressed in Yoon et al (2018) <4>.
>           The authors maintained the practice of freezing old weights in old
>          tasks, but they only introduced the new weights when some
>          predetermined threshold was met (through an iterative algorithm see
>          Algorithm 1 in <4>).
>      A similar network compartmentalisation to <2> - <4> can be found in Ebrahimi
>      et al (2020) <5>,
>          where the authors factorized the latent space into shared and private
>          parts. And that a new private part was learnt per task and frozen (see
>          P^k in Equation (3) of <5>).
>      Last, a near-identical network freezing compartmentalisation can be found in
>      Lomonaco et al (2020) <6>
>          along with their proposed CWR+ method.
>
>
>  	<1>: Nguyen et al (2020).
>  		Dissecting Catastrophic Forgetting in Continual Learning by Deep
>  		Visualisation
>  	<2>: Golkar et al (2019).
>  		Continual Learning via Neural Pruning
>  	<3>: Rusu et al (2016).
>  		Progressive Neural Networks
>  	<4>: Yoon et al (2018).
>  		Lifelong Learning with Dynamically Expandable Networks
>  	<5>: Ebrahimi et al (2020).
>  		Adversarial Continual Learning
>  	<6>: Lomonaco et al (2020).
>  		Rehearsal-Free Continual Learning over Small Non-IID Batches
>
>
> <<<>>>
>
> Our backbone network designs are different for Perm-MNIST,
> and for Inc-Cifar100 and Inc-CUB200 (see “The Backbone Networks”
> on page 6 of our manuscript). The main reason is because Cifar100 and CUB200 are both much more contextually rich than the MNIST dataset.
> Due to this reason, fixed feature extractors are introduced for Cifar100 and
> CUB200 but not for MNIST. However, the overall modifications to the classifier networks are not big. You can find additional illustrations in Figures 7 and 8 in Appendix B on page 14 of our manuscript.

---

> ### Author Response · Authors · 2021-11-17
> **Thread # 2 to Reviewer Homk**
>
> With regards to your comment:
> Comparison with relevant work is missing: The idea of sparse, selective update of parameters to mitigate catastrophic forgetting and encourage task separability has been explored previously. For example, [1] with energy-based models, [2] with local synaptic plasticity and neuromodulation, [3] learns relevance maps. The UCL approach was also mentioned in the paper but not compared against. A systematic comparison with this class of approaches will help to better assess the proposed approach in terms of accuracy/computational cost.
> <<<>>>
>
> We thank the reviewer for suggesting these three works.
>      [1]: Li et al (2021)’s energy-based model,
>      [2]: Madireddy et al (2021)’s neuromodulation, and
>      [3]: Kausik et al (2021)’s relevance map.
> However, we would like to point out that all of these methods cannot support generic and modular network modification to achieve continual learning. Especially, Kausik et al requires the knowledge of task boundary while our algorithm, as well as chosen baselines and SOTA, work without such knowledge.
>
> The other 4 reviewers all agree that our experiments are comprehensive. With Reviewers zxFC, pQdx, and tojU finding our experiments convincing; and with Reviewer h7Ku finding our experiments interesting. We consider the main advantage of our method in its simplicity. This was nicely summarised by Reviewer tojU:
> “…Though the proposed modifications to the HCN architecture are rather simple, I
> think this paper is worthy of note, because these changes make the network quite
> more robust for continual learning, and thus present a relatively straight forward,
> headache free (other than deciding on the MHC threshold) architecture that
> seems significantly less vulnerable to catastrophic forgetting.”
>
> Nonetheless, we have conducted some extra experiments on Li et al (2021)’s Energy-Based Models (EMBs) to demonstrate the effectiveness of our method.
> +/-: confidence interval (1.96 * standard deviation)
>
> For     Perm-MNIST:
>
>  	Ours         	Classifier             	ACC (%)         	FA1(%)
> 				EBM                 	74.19             	61.09 +/- 13.68
>          			EBM + HCN             77.66             	65.79 +/- 11.07
>  	Yes         	EBM + MHC(70%)   	77.94             	66.80 +/- 14.41
>  	Yes         	EBM + LWN             80.32             	74.93 +/- 6.10
>  	Yes         	EBM + LWN in HCN 	82.26             	77.04 +/- 4.20
>  	Yes         	EBM + LWN             82.10          	 	76.65 +/- 2.38
>  				in MHC (70%)
>
> For     Inc-Cifar100:
>
>  	Ours         	Classifier             	ACC (%)         	FA1(%)
>          			EBM                 	31.30             	21.78 +/- 9.40
>  	Yes         	EBM + LWN             37.44           	31.61 +/- 3.22
>  				in MHC (10%)
>
> For     Inc-CUB200:
>
> 	Ours         	Classifier             	ACC (%)         	FA1(%)
>  				EBM                 	32.38             	15.15 +/- 4.69
>  	Yes         	EBM + LWN             51.32           	30.76 +/- 5.53
>  				in MHC (10%)
>
> Note that in the original EBM paper, Li et al (2021) only reported their model performances in ACC. In addition, they did not conduct experiments over the Inc-CUB200 setting. Further note, that our baseline results were similar to theirs, as reported in Table 1 in page 5 of their preprint paper
> (https://arxiv.org/pdf/2011.12216.pdf).
> All of these new results and their experimental setups will be addressed as a new appendix in our revised manuscript.
>
> In order to further discuss the papers that you mentioned, we will summarise those 3 papers that you mentioned in the next thread.

---

> ### Author Response · Authors · 2021-11-17
> **Thread # 3 to Reviewer Homk**
>
> This thread is an appendix to Thread # 2
>
> <<<>>>
>
> In their paper, Li et al (2021) argued that softmax classifiers encouraged
> the backbone network to learn a winner-takes-all dynamic. Hence,
> they exchanged the conventional softmax classifier with an energy-based model.
> As shown in Section 4.3.2 on the right column of page 4 of their paper, a slight drawback to their method is that they required the extra information on the input-target pair.
> This could also be found in the official implementation of Li et al in Line 29 of
> https://github.com/ShuangLI59/ebm-continual-learning/blob/531aaa6c2a6d64ebcbb407c1a053aaf921e43159/network/ebm_layers.py
>
> We consider the contributions of our techniques (MHC and LWN) to be orthogonal to the contribution of Li et al. On one hand, our contribution focuses on employing the right intermediate layers for the backbone network; while on the other hand, Li et al’s paper focused on choosing a better formulation for the last layer. Due to the dissimilarity in the purposes, Li et al’s method was not initially considered as a baseline for our manuscript. Furthermore, the dissimilarity in our modifications also means that our algorithms can be implemented alongside with Li et al’s energy-based method. Please find the results in Thread 2.
>
> <<<>>>
> Madireddy et al (2021)’s paper introduced a biologically-inspired
> meta-continual learning technique that transfers learnt knowledge quickly
> without memory intensive replay. In Section 3.4 on page 4 of their paper, Madireddy et al added a built in random forest approach and thus required an additional sleep-wake cycle.
>
> Their method thus has a paradigmatic shift from convention CL methods.
> This is thus orthogonal to the goal of our paper; of which is to find simple techniques
> that can be built upon most if not all existing methods and help them to achieve
> better results.
>
> <<<>>>
>
> Kaushik et al (2021)’s relevance mapping aims to assign larger weights to
> more important synaptic connections in their backbone network. However,
> this paper invovles 2 significant caveats that demand attention.
>
> First and as shown in Equation (7) on the left column on page 7 of their paper,
> their mask explicitly defines a predetermined structure for their backbone network. This is comparatively more complicated than our MHC and Kuo et al (2021b)’s
> HCN design. Second and arguably more undesirable, their method requires the learning of a separate mask “per task”. Note that on page 7 of their paper, the fact that each map is denoted as M_{p_k} for each task k means that the task boundary needs to be known. Sequential learning with task boundary known is a significant and unfair advantage feature that many recent continual learning papers are aiming to remove.
>
> <<<>>>
>
> Furthermore, it was suggested that we should
> compare our method with Ahn et al (2019)’s UCL method
> because we mention it in our paper.
> However, like the influential EWC (Kirpatrick et al, 2017),
> UCL is a regularisation-based continual learning method.
> We have explicitly stated in our introduction (paragraph 2 on page 1 of our manuscript) that we aim to remove all external means, including regularisation,
> to retain learnt knowledge.
>
> In addition, our experiments in Table 1 on page 7 of our manuscript have
> already compare our modifications to EWC. We have therefore compared
> our methods to existing regularisation-based continual learning approaches.

---

> ### Author Response · Authors · 2021-11-17
> **Thread # 4 to Reviewer Homk**
>
> Regarding the comment:
> Continual learning approaches are typically evaluated in on task- and class-incremental learning settings, it’s not clear from the experiments about which scenarios is considered.
>
> <<<>>>
>
> The methods in our paper are evaluated on the popular domain-incremental learning (Van de Ven et al, 2019)<1> which we solved tasks that we have observed so far without any of the task-ID provided.
> We will clarify this in our final manuscript.
>
> The main reason why we choose to evaluate our methods
> on this setting is because that our 4 baselines,
>
>      HCN (Kuo et al, 2021b),
>      EWC (Kirpatrick et al, 2017),
>      ER (Chaudry et al, 2019), and
>      HAT (serra et al, 2018)
>
> were all tested on the domain-incremental learning setting in their
> respective original papers.
>
> It should be further note that, our methods (MHC and LWN) are proposed as
> simple “patches” to help existing continual learning techniques
> to achieve even better results from their current form. Hence, we think that
> it is necessary for us to keep the identical experimental settings as
> these prior work.

---

> ### Author Response · Authors · 2021-11-17
> **Thread # 5 to Reviewer Homk**
>
> Regarding the comment:
> Classification accuracy might not be the best metrics to evaluate the continual learning potential. Other metrics such as the backward and forward transfer are better suited [4].
>
> <<<>>>
>
> In this very recent survey paper by Mai et al (2021) that you mentioned,
> Mai et al also acknowledged on Section 9 in page 17 and Section 10 in page 18
> that many papers also have not supplied information on backward transfer.
> We think that it is probably because this isn’t adopted as a standard practice yet.
> If fact, the 3 papers that you mentioned
>
>    	[1]: Li et al (2021)’s energy-based model,
>      	[2]: Madireddy et al (2021)’s neuromodulation, and
>      	[3]: Kausik et al (2021)’s relevance map;
>
> are all from 2021, and that all of them only reported accuracy as their only evaluation metrics.
>
> In addition, it should be noted that there are 2 metrics in our paper.
> See page 6 in our manuscript, all methods in our work are evaluated with
>
>   	the average accuracy (ACC) and
>      	the final accuracy of Task 1 (FA1).
>
> FA1 can hence be seen as a proxy for backward transfer because it reflects
> the final performance of the backbone network. In addition, FA1 has one
> additional advantage over backward transfer and that it can reflect how
> severely overfitted a backbone network is to the earlier tasks
> in the continual learning setup. We hence consider our experiments to be
> relatively thorough.

---

> ### Author Response · Authors · 2021-11-17
> **Thread # 6 to Reviewer Homk**
>
> Your comment:
> Do the results assume a certain task ordering or have they been chosen randomly. Does the accuracy change with the change in relative ordering of the tasks? Will having consecutive having tasks with similar distributional properties have any effect on the interference with previous tasks.
>
> <<<>>>
>
> Three things should be noted and considered together to answer this.
>
>  	First as mentioned in pages 5 and 6 in our manuscript,
>  		our backbone networks run over 10 randomly initialised seeds and
>           	we report the average result with 95% confidence interval (+/- 1.96
>          	standard deviation).
>    	Second, all of the tasks for Perm-MNIST are randomised;
>          	while we used Douillard et al (2020)’s code for the Inc-Cifar100 and
>          	Inc-CUB200. Douillard et al’s code is popular among recent continual
>          	learning papers.
>     	Third, for Inc-Cifar100 and Inc-CUB200, the fixed feature extractor were
>          	pre-trained on ImageNet.
>
> We will discuss how these three aspects interplay below.
>
> We use the codes from Douillard et al to compare with previous state of the arts
> (mentioned in footnote 2 on page 6 of our manuscript). However, it should be noted that his code
> (see line 51 and 294 of https://github.com/arthurdouillard/incremental_learning.pytorch/blob/master/inclearn/lib/data/datasets.py)
> explicitly defines the orders of the classes; there can therefore be some similar distributional properties. This could be further complicated by any contextual similarity between the contents of ImageNet to Cifar100, and of ImageNet to CUB200.
>
> However, these are all out of the scope of our paper. Our implementation follows the settings of previous papers and we have carefully documented them in our manuscript. Furthermore, our metric takes aggregated the results over 10 random seeds; and thus the significance of individual tasks (due to ordering) is scaled down.

---

### Author Response · Authors · 2021-11-17
**A General Comment**

The authors of this manuscript would like to thank all reviewers for your reviews.
We have now provided our replies to your respective concerns. Below, we address some general changes that we have made to our revised manuscript.

We have submitted a rebuttal revision of our manuscript. Specifically, we made 3 modifications.


     <<<>>>
     (1)     Reviewer tojU suggested us to further clarify the relationship between
          gradient vanishing and catastrophic forgetting.
           To this end, we added underbraces and overbraces to Equation (3)
         on page 3 in our manuscript to highlight the specific functionality of
         each term and their individual relations to gradient vanishing and
         catastrophic forgetting.
     <<<>>>
     (2)     Reviewer zxFC suggested us to supplement Figure 3 (on page 5) and
         Figure 4 (on page 8) with the exact amount of changes that occurred to
         the learnt mapping of MNIST after the classifiers started to observe
         data from FashionMNIST.
         To this end, we computed the exact amount of change that occurred to
         the learnt mappings and quantify the amount of forgetting that could be
         mitigated with our LWN technique.
         In order to make space for this new description, we moved Figure 3(c)
         from the main text to Appendix B on page 13 of our manuscript.
      <<<>>>
     (3)     Reviewer Homk suggested us to conduct more experiments; and
         especially to compare our techniques to classifiers that do not
         utilise the soft max layer.
         To this end, we added a new set of experiments in Appendix F on
         page 19 of our manuscript. In that appendix, we compared our MHC
         and LWN to Li et al (2020)’s backbone networks which employed
         energy-based models (EBMs) for continual learning. Our results
         showed that both MHC and LWN were compatible to EBMs and
         helped it to achieve better performances on 3 different datasets.

We also made some minor changes to the revised manuscript to improve the readability and to adjust some grammatical errors.

---

### Decision · Program_Chairs · 2022-01-20

**Decision:**

Reject

**Comment:**

Unfortunately, I feel the paper is not quite ready for ICLR, even if the reviews seem in general quite positive (though of low confidence).
After reading the reviews and rebuttal, and going over the paper I have to make the following comments:
 * The paper do two modification to the backbone architecture, that have an impact on the ability of these systems to continually learn; these changes are adding layer normalization and a mask
  * The paper is mostly empirical in nature; while there are some intuitions presented clearly about these ideas, their efficiency is proved empirically, which is completely fine

 However:
  * The empirical validation seems not sufficient; the main results are permuted MNIST, incremental CIFAR 10, incremental CUB200; the results on permuted MNIST in terms of final accuracy seem surprisingly low (particularly when involving CL solutions, like EWC, ER, HAT .. see table 2; e.g. FA1 < 80% seems very surprising). This seems strange to me and adds a bit of shade on the results
  * The proposed methods are simple; There is a strong message behind them, namely that the choice of the backbone (architecture size, normalization layers) has a huge impact on learning. But being a purely empirical result, this really needs to be backed up with analysis and an attempt at understanding of what is going on. E.g. looking at the masks over time .. to they converge to be task specific? Anything that would give a bit of depth to the results. Discussing the Figures (e.g. I'm looking at Fig 3 and I was grep-ing the text to see a discussion of how one would interpret those results). Why is FashionMNIST used to produced Fig 3, and why is not something like this done for one of the CL benchmark considered. Providing additional typical measures for CL (e.g. showing learning curves).
  * Just overall does not seem that the work provides sufficient insight, or analysis.

I do think there is something really interesting in this work, and I do hope the authors will resubmit this work after some modification. And I do agree that there are many aspects of the backbone or architecture that have big impacts on CL and this is an understudied and not well understood topic. So in that sense I think the idea of this work is good. But I just feel it fails short in terms of results, analysis. I feel in the current format, the work will not have the impact it deserves.